# FUG: Feature-Universal Graph Contrastive Pre-training for Graphs with Diverse Node Features

**Jitao Zhao[1], Di Jin[1],**\* **Meng Ge[2], Lianze Shan[1], Xin Wang[1], Dongxiao He[1], Zhiyong Feng[1]**

[1]College of Intelligence and Computing, Tianjin University, China
[2]Department of Electrical and Computer Engineering, National University of Singapore, Singapore
[1]{zjtao, jindi, shanlz2119, wangx, hedongxiao, zyfeng}@tju.edu.cn, [2]gemeng@nus.edu.sg

## Abstract

Graph Neural Networks (GNNs), known for their effective graph encoding, are extensively used across various fields. Graph self-supervised pre-training, which trains GNN encoders without manual labels to generate high-quality graph representations, has garnered widespread attention. However, due to the inherent complex characteristics in graphs, GNNs encoders pre-trained on one dataset struggle to directly adapt to others that have different node feature shapes. This typically necessitates either model rebuilding or data alignment. The former results in non-transferability as each dataset need to rebuild a new model, while the latter brings serious knowledge loss since it forces features into a uniform shape by preprocessing such as Principal Component Analysis (PCA). To address this challenge, we propose a new Feature-Universal Graph contrastive pre-training strategy (FUG) that naturally avoids the need for model rebuilding and data reshaping. Specifically, inspired by discussions in existing work on the relationship between contrastive Learning and PCA, we conducted a theoretical analysis and discovered that PCA's optimization objective is a special case of that in contrastive Learning. We designed an encoder with contrastive constraints to emulate PCA's generation of basis transformation matrix, which is utilized to losslessly adapt features in different datasets. Furthermore, we introduced a global uniformity constraint to replace negative sampling, reducing the time complexity from $O(n^2)$ to $O(n)$, and by explicitly defining positive samples, FUG avoids the substantial memory requirements of data augmentation. In cross domain experiments, FUG has a performance close to the re-trained new models. The source code is available at: https://github.com/hedongxiao-tju/FUG.

## 1 Introduction

Graph Neural Networks (GNNs) [1], renowned for their effectiveness in encoding both graph topology and feature information, have emerged as the mainstream methods in graph representation learning. They are extensively applied in various scenarios such as protein molecule prediction [2], e-commerce recommendation systems [3], and community detection [4]. Training high-quality GNNs typically relies on extensive manually labeled data [5], which is costly and limits their applications. To address this challenge, graph self-supervised pre-trainings have gained significant attentions [6, 7, 5]. These approaches leverage self-supervised signals mined from the graph data themselves and design proxy tasks to train models without manually labeled data.

Unfortunately, due to the inherent complexity of graph data, models trained on one graph dataset often fail to perform well on another. This complexity arises from two main aspects: 1). topological

---

\*Corresponding author

38th Conference on Neural Information Processing Systems (NeurIPS 2024).

diversity, where a node in a graph can connect to any number of other nodes, and 2). feature diversity, where nodes represent a variety of information such as textual or float data. In domains like vision [8], text [9], and speech [10], data shares common latent semantics (such as RGB channels in pixels, word vectors, or audio samples), and have relative positional information like spatial relations in vision or sequence relations in text and audio. Unlike these domains, features in graphs are unordered and chaotic. Even within the same citation dataset category using identical feature extraction methods, such as Cora, CiteSeer, and PubMed [11, 12], the node features have different shapes.

Existing GNNs can address the issue of topological diversity through well-designed message-passing and aggregation methods [13, 14]. However, due to the neural networks' requirement for fixed input-output dimensions, GNNs struggle with feature diversity. This necessitates retraining a new graph neural network for datasets with different node feature shapes, significantly restricting the applicability of existing graph self-supervised pre-training models. This leads to some graph pre-training works [6, 15] that have to ignore node attributes and encode only the structural information. Recent "All-In-One" work in graph attempts to reshape node features to the same dimensions through preprocessing methods like Principal Component Analysis (PCA) [16, 17], feature similarity embedding [18] and language models [17]. However, these methods inevitably lose crucial semantic information related to downstream tasks during data preprocessing. Classical machine learning methods such as PCA cannot participate in training, and they primarily encode the information with the highest variance as principal components, overly focusing on differences between node features and losing substantial homophily information in node features. Encoding informative features into single-dimensional similarity also brings obvious information loss. Language models suffer assumption biases, presuming that all input data is textual, thus inef-

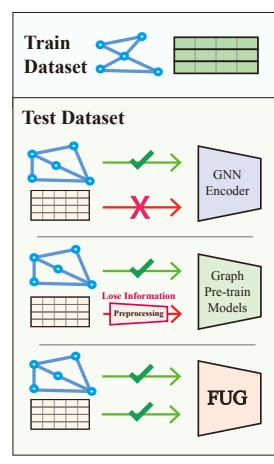

Figure 1: Motivation.

fectively handling extensive float features. Furthermore, using large language model interfaces or involving language models in training incurs additional significant costs. Additionally, sliding window-based neural networks, such as 1D-CNNs [19] and LSTMs [20], also fail to address the issue of feature diversity. While they can handle inputs of varying dimensions, they operate under the prior assumption that the data is sequential, which is not satisfied by graph node features.

Among the feature-unifying approaches discussed earlier, only PCA-like methods exhibit no assumption bias and are adaptable to various feature forms. Their primary limitation lies in the misalignment between their model objectives and the proxy tasks of graph pre-training, coupled with their inability to train based on specific loss functions. Noting discussions in existing works that link contrastive learning with PCA-like methods, which regard contrastive pre-training as a generalized form of non-linear PCA [21], we draw inspiration to propose a question: *Could a graph contrastive pre-training principle based on PCA theory be developed to address inherent issues of PCA and broadly applied to node features of different shapes?*

In this paper, we theoretically demonstrate that classical PCA and contrastive learning fundamentally follow the same theoretical framework. PCA's optimization objective is a special case of that in contrastive learning. The adaptability of PCA-like methods to various feature shapes stems from their intrinsic component that acts as a columnar encoder for values under specific feature dimension, creating a basis transformation matrix, which we term dimensional encoding. The loss and parameter update mechanisms of contrastive learning could offer traditional PCA methods new goals and training mechanisms that align better with the graph data and downstream tasks. Based on these analyses, we propose a new strategy for **F**eature-**U**niversal **G**raph pre-training (FUG strategy), which includes three steps: dimensional encoding, graph encoding, and relative relationship optimization task.

Guided by this theory, we instantiate a simple FUG model. Simulating PCA, this model utilizes sampling to capture the distribution among different nodes within specific feature dimension for dimensional encoding, and generates basis transformation vectors for feature dimensions which are used to losslessly unify the node features. It employs GNN encoder for graph encoding. We also design an improved and more efficient graph contrastive loss as the relative relationship optimization task. According to this task, FUG effectively learns the relative relations within graph data, and transfers this encoding capability effectively to other datasets with different node feature shapes.

We conducted extensive experiments to validate FUG's efficacy. In in-domain self-supervised learning experiments, FUG demonstrated competitive performance with existing advanced graph self-supervised models. In the cross-domain learning experiments, the FUG trained and tested on different datasets shows similar performance to the model trained and tested on the same dataset. These results prove the strong adaptability and transferability of FUG.

## 2 Notations and Preliminary

In this section, we introduce some preliminary concepts and notations to facilitate analysis in later. We consider a graph data $\mathcal{G} = (\mathcal{V}, \mathcal{E})$, where $\mathcal{V} = \{v_1, v_2, ..., v_n\}$ represents the set of nodes and $E \subseteq V \times V$ represents the set of edges. $X \in \mathbb{R}^{n \times d}$ denotes the feature matrix of the nodes. $A$ is the adjacency matrix representing the topological relationships between nodes, with $a_{ij} = 1$ if $e_{ij} \in \mathcal{E}$ and $a_{ij} = 0$ otherwise.

**Graph Neural Networks (GNNs)** . GNNs can be uniformly described as $H^{(t+1)} = \widetilde{A} W H^{(t)}$, where $H$ represents the embeddings in hidden layer, with $t = 0$ denoting the node features $X$. $\widetilde{A}$ is the message passing matrix, which can be derived through neighbor sampling [13], random walks, or deep learning [22] techniques to mine graph data. $W$ is a learnable parameter matrix. Under the successful design of existing works, the message passing matrix $\widetilde{A}$ can generically encode graph data with varying edges. However, the presence of the parameter matrix $W$ means that a trained GNN model can only be directly applied to graphs with the same node feature dimensionality as the training dataset, significantly limiting the application scenarios of GNNs.

**Graph Contrastive Pre-training**. It is based on Contrastive Learning (CL). Given graph $\mathcal{G}$, graph contrastive pre-training aims to train a GNN encoder $f(\cdot)$ in a self-supervised manner, which generates low-dimensional dense graph representations $Z = f(\mathcal{G})$ for downstream tasks. Typically, InfoNCE loss functions are used for training, formulated as:

$$\mathcal{L}_{\text{InfoNCE}} = -\log\left(\frac{\sum_{(v_i,v_j)\in PS} e^{\text{Sim}(g(v_i),g(v_j))/\tau}}{\sum_{(v_i,v_j)\in PS} e^{\text{Sim}(g(v_i),g(v_j))/\tau} + \sum_{(v_i,v_j)\in NS} e^{\text{Sim}(g(v_i),g(v_j))/\tau}}\right), \quad (1)$$

where $PS$ denotes the data sampled as positive pairs, and $NS$ denotes the data sampled as negative pairs. $\text{Sim}(\cdot)$ represents the cosine similarity calculation, $g(\cdot)$ is a non-linear projector, $\tau$ is a temperature parameter. The training objective is to increase the similarity between positive pairs and decrease the similarity between negative pairs.

**Principal Component Analysis (PCA)**. Given data $X$ with any number of dimensions $d$, PCA aims to reduce the dimensionality to $k$ dimensions, where $k \leq d$, while maximizing the retention of variance between the data. PCA based on eigendecomposition is shown as:

$$\text{PCA}(X) = \widetilde{X}P, P = \text{Norm}[\text{Concat}\{c_i|c_i \in C, \lambda_c \geq \lambda^{(k)}\}], (\Lambda, C) = \delta[\text{Cov}(\widetilde{X_{:,1}}, \widetilde{X_{:,2}}, ..., \widetilde{X_{:,d}})], \quad (2)$$

where $P$ represents the basis transformation matrix, $\widetilde{X} = X - \text{Mean}(X)$. $c_i$ denotes the $i$-th eigenvector, $\lambda_i$ denotes the corresponding eigenvalue, $\lambda^{(k)}$ denotes the $k$-th largest eigenvalue. $\text{Concat}(\cdot)$ represents concatenating the eigenvectors within the set, and $\text{Norm}(\cdot)$ represents normalization. $C$ and $\Lambda$ represent the matrix composed of eigenvectors and the corresponding eigenvalues, respectively. $\text{Cov}(\cdot)$ represents covariance computation, and $\delta$ denotes matrix decomposition. Note that since the covariance matrix is symmetric, the extracted eigenvectors are orthogonal to each other. This formulation mathematically ensures that the dimensionality-reduced embeddings maximize the retention of variance between the original data and minimize homogeneity among the data.

## 3 Graph Contrastive Pretrained Models from the PCA perspective

In this section, we first theoretically summarize the framework that PCA essentially follows. We then analyze contrastive learning, demonstrating that it also adheres to this framework, and the PCA optimization task is a special case of that of CL. Furthermore, we answer the question why the PCA embeddings generally underperform graph contrastive pre-training in downstream tasks within the same framework, and why graph contrastive pre-training models lose the ability to adapt datasets with varying node feature shapes. Based on these findings, we propose a new strategy for Feature-Universal Graph pre-training (FUG strategy). First we define a class of objectives:

**Definition 3.1 (Data relative relationship optimization task $\mathcal{L}_{RT}$)** . *Given a dataset $X$ and a model $\phi(\cdot)$, $\mathcal{L}_{RT}$ involves maximizing or minimizing the $l2-$distance between certain pairs of data points, formulated as:*

$$\text{Minimize } [\mathcal{L}_{RT}] \propto \text{Minimize } [\mathbb{E}_{(x_i,x_j) \in X^-}[Sim(z_i, z_j)] - \mathbb{E}_{(x_i,x_j) \in X^+}[Sim(z_i, z_j)]], \quad (3)$$

*where $X^+$ represents the set of data pairs that need to be encoded as similar, whereas $X^-$ denotes the set of pairs that should be encoded as dissimilar. And $z_i$ denotes the embedding of $x_i$.*

Notably, unlike the contrastive loss, $X^+$ represents positive pairs within the raw data, rather than positive pairs between the augmented views. It means $\mathcal{L}_{RT}$ focuses solely on the relative semantic relationships among the data points, disregarding their absolute semantic.

**Definition 3.2 (Encoder Dim$(\cdot)$)** *Dim$(\cdot)$ refers to a series of dimension-specific encoders that directly take the data of all samples under a specific dimension $i$, $X_{:,i}^T$, and output the embedding of that dimension, $DimEmb_i$, formalized as $DimEmb_i = Encode(X_{:,i}^T)$, $DimEmb \in \mathbb{R}^{D \times N'}$.*

**Definition 3.3 (Encoder Fea$(\cdot)$)** *Fea$(\cdot)$ refers to a series of sample-specific encoders that take the data of all dimensions of a specific sample $j$, $X_j$, and output the embedding of that sample, $FeaEmb_j$, formalized as $FeaEmb_j = Encode(X_j, \epsilon)$, $FeaEmb \in \mathbb{R}^{N \times D'}$, where $\epsilon$ represents additional input.*

**Theorem 3.1 (PCA framework)** . *Consider $Dim(\cdot)$ as an encoder modeling the distribution of data across a specific dimension (as in Definition 3.2), $Fea(\cdot)$ as an encoder modeling the relationships between different dimensions of data (as in Definition 3.3). PCA is an instantiation model for these two encoders. Assuming that data representations have equal length $m$, when $X^+ = \emptyset$ in Definition 3.1 of $\mathcal{L}_{RT}$, PCA adheres to the following form:*

$$\text{Minimize } [\mathcal{L}_{PCA}(PCA(X))] \propto \text{Minimize } [\mathcal{L}_{RT}(Fea(Dim(X)))]. \quad (4)$$

Rigorous proof can be found in Appendix A.1. Here, $\mathcal{L}_{PCA} = -\mathbb{E}[||PCA(x_i) - \text{mean}(PCA(X))||_2^2]$ is interpreted as an instantiation of $\mathcal{L}_{RT}$, based on the principle of maximizing the retention of data variance. We consider the covariance function in Eq. 2, $Cov(\cdot)$ , as an implementation of $Dim(\cdot)$, since it essentially encodes the distribution information of a single dimension through the computation of covariance. The matrix decomposition and normalization concatenation in Eq. 2 are viewed as an implementation of $Fea(\cdot)$, as they fundamentally encode the relationships among different data dimensions. This interpretation also applies to PCA methods based on Singular Value Decomposition. Thus, we summarize the framework that PCA adheres to as shown in Eq. 4.

**PCA framework has powerful transfer ability.** From Eq. 4, it can be inferred that PCA essentially functions as a neural network model trained to reach a global optimum. The parameters of PCA's two encoders learn to encode the covariance of dimensional data and to extract and concatenate eigenvectors to form the basis transformation matrix, respectively. Extensive researchers have validated PCA's success across various fields with the same parameters [23, 24, 25], demonstrating that under the guidance of the $\mathcal{L}_{RT}$ loss, the model's capacity to encode difference in data demonstrates formidable transferability. This transform capability originates from the model's focus on encoding the differential relationships among data rather than their absolute semantics.

PCA is capable of adapting to arbitrary Euclidean data and demonstrates strong transfer performance. However, its effectiveness generally falls short of advanced NN-based models, and it lacks the ability to encode structure information. To address the issue of node feature generality in graph pre-training models, inspired by existing work [26], we attempt to further theoretically analyze the relationship between existing graph contrast pre-training models and the PCA framework. We start by analyzing the InfoNCE loss which is commonly used in contrastive learning.

**Theorem 3.2 (Equivalence of Contrastive Loss and $\mathcal{L}_{RT}$)** . *Assuming that data augmentation in contrastive learning does not alter the essential semantics of node pairs for downstream tasks, it follows:*

$$\text{Minimize } [\mathcal{L}_{InfoNCE}] \propto \text{Minimize } [\mathcal{L}_{RT}]. \quad (5)$$

Detailed proof can be found in Appendix A.2. Here we offer an intuitive explanation: the negative sampling component in InfoNCE loss is widely acknowledged for encoding dissimilarities within

the data. We contend that the augmentation-based positive sampling part, inherently encodes similar relationship between data points rather than their absolute semantics. Consider a pair of data points, $x_i$ and $x_j$, with the same semantics. Ideally, when data augmentation does not alter the essential semantics of the nodes, the augmented data, $Aug(x_i)$ and $Aug(x_j)$, should be identical. Consequently, Maximize $[\text{Sim}(x_i, Aug(x_i)) + \text{Sim}(x_j, Aug(x_j))]$, essentially equates to Maximize $[\text{Sim}(x_i, x_j)]$. Thus, data augmentation is essentially looking for positive samples in the dataset, aligning with the views expressed in work [27]. Furthermore, this view explains why augmentation is less critical in graph contrastive learning compared to other domains [28], leading to the proposal of many augmentation-free graph contrastive studies [29, 30, 31], as the edges in graph data (especially in homophilic graphs) inherently convey similarities between data points.

**Corollary 3.2.1** *Consider a specific instantiation of a dimension encoder defined as $Dim(X) = X$, and the contrastive model viewed as an instantiation of an $Fea(\cdot) = ContrastiveModel(\cdot)$. Under this the contrastive model also adheres to the framework in Theorem 3.1, which states:*

$$\text{Minimize } [\mathcal{L}_{InfoNCE}(ContrastiveModel(X))] \propto \text{Minimize } [\mathcal{L}_{RT}(Fea(Dim(X)))]. \tag{6}$$

From this, we can conclude that the PCA optimization objective is, in fact, a special case of the CL loss, detailed proofs are in Appendix A.2. As demonstrated by Theorem 3.1 and Corollary 3.2.1, PCA and contrastive pre-training fundamentally follow the same framework. Our analysis extends the existing discussions [26] on the relationship between the two. Within this framework, we can more clearly analyze why PCA is versatile across various data, while advanced contrastive pre-training models generally outperform PCA when directly used as embeddings in downstream tasks.

**Answer 1. The robust versatility of PCA across various types of Euclidean data is attributed to its inclusion of the Dim$(\cdot)$ component.** As illustrated in Table 1, PCA, in contrast to contrastive learning models, incorporates the Dim$(\cdot)$ component. Most existing neural network layers encode features of individual data points rather than encoding across the same dimension for different data. Due to these layers' fixed input-output dimensionality requirements, they cannot universally adapt to data with varied feature forms except using a sliding window

Table 1: Comparative Analysis of Model Components in PCA, Contrastive Pre-training Models (CL), and the proposed FUG

|  | Dim$(\cdot)$ | Fea$(\cdot)$ | $X^+$ | $X^-$ |
|---|---|---|---|---|
| PCA | ✓ | ✓ | ✗ | ✓ |
| CL | ✗ | ✓ | ✓ | ✓ |
| FUG (Ours) | ✓ | ✓ | ✓ | ✓ |

strategy. In contrast, PCA encodes different data within the same dimension by computing covariance between dimensions, inherently encoding the data distribution of that dimension. This enables PCA to generate transform matrix and integrate data into embeddings within a unified shape.

**Answer 2. The superior performance of contrastive pre-training models over PCA primarily stems from their additional encoding of data homogeneity and a more flexible Fea$(\cdot)$ component.** As noted in Table 1, a key advantage of contrastive learning (CL) compared to PCA is its consideration of homogeneity in the optimization objective, rather than merely encoding differences between data points. Furthermore, thanks to a variety of neural network layers, these models are capable of extracting richer data information and are adaptable to non-Euclidean graph data.

**Feature-Universal Graph Contrastive Pre-training.** Based on the above analysis, we propose that a Feature-Universal Graph Contrastive Pre-training model should include a dimension-specific data distribution encoder Dim$(\cdot)$, a cross-dimensional data information encoder Fea$(\cdot)$, and an optimization objective $\mathcal{L}_{RT}$ that encodes both similarities and differences between data. Models designed under this theory can universally adapt to graph data with any node feature shapes. Moreover, guided by $\mathcal{L}_{RT}$, the model captures the relative semantic relationships among data, rather than absolute semantics. This type of knowledge, which is more easily transferable, ensures that the model can generalize across different datasets.

## 4 Feature-Universal Graph Pre-training Model

Guided by the theories presented in Section 3, we instantiated a Feature-Universal Graph pre-training model (FUG model), as depicted in Fig. 2. This model incorporates a Dimension Encoding component DE$(\cdot)$, a Graph Encoder GE$(\cdot)$, and a relative relationship optimization task $\mathcal{L}_{\text{RT-FUG}}$. The DE$(\cdot)$

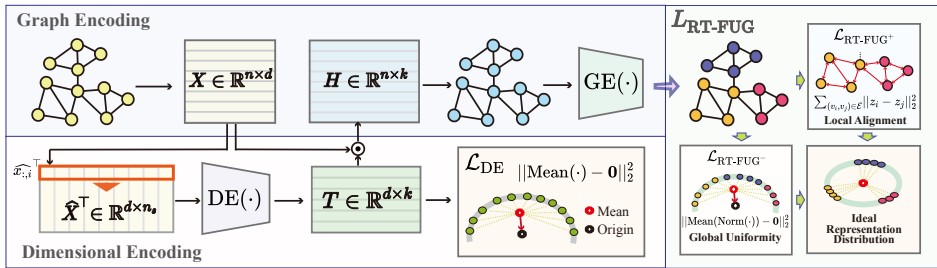

Figure 2: The overview of the proposed FUG model. Given the node features $X$ and structure $A$ of a graph, $X$ is first processed through a learnable dimensional encoding component, $DE(\cdot)$, to generate a basis transformation matrix $T$, which is used to embed features into a unified shape $H = XT \in \mathbb{R}^{n \times k}$. $H$ and $A$ are then input into the graph encoder $GE(\cdot)$ to produce representations $Z$. We set three losses, $L_{DE}$, $L_{RT\text{-}FUG^+}$, and $L_{RT\text{-}FUG^-}$, which collaboratively train $GE(\cdot)$ and $DE(\cdot)$.

learns the data distribution within a specific dimension to generate a basis transformation matrix $T$, which converts any form of node features to a unified shape. The $GE(\cdot)$ encodes both the structure and the node features to obtain graph representations. $\mathcal{L}_{RT\text{-}FUG}$ is an improved contrastive loss based on the $\mathcal{L}_{RT}$ concept. Compared to the classical InfoNCE loss, $\mathcal{L}_{RT\text{-}FUG}$ more effectively guides the model in encoding the relative semantic information in the data. Furthermore, in line with the earlier discussion on augmentation essentially serving as positive sampling, we explicitly define positive sampling within $\mathcal{L}_{RT\text{-}FUG}$ to replace data augmentation.

**Dimension Encoder DE$(\cdot)$.** We designed a non-linear dimension encoder, $DE(\cdot)$, to encode the distribution information of a specific dimension. If it encodes the features of all nodes under one dimension, with the input dimension fixed at the number of nodes in the training set $n_t$, this encoder would be constrained by the number of input nodes. Therefore, we randomly sample $n_s$ nodes from the training set as the input for this encoder, approximating the feature distribution of the entire data through the sampled nodes. The dimension distribution information, after being processed by $DE(\cdot)$, generates a basis transformation vector for each dimension as follows:

$$t_i = DE(X_{:,i}) = \text{Norm}[\text{MLP}(\widehat{X_{:,i}}^\top)], \tag{7}$$

where $\widehat{X} = \text{Sample}(X)$ and MLP represents a Multi-Layer Perceptron. $X_{:,i}$ represents the i-th column of the X matrix, that is, all the data under the i-th feature dimension. Eq. 7 can be seen as $DE(\cdot)$ assigning a point to each feature dimension in a hyperspherical embedding space which the center at the origin and radius one. The basis transformation vectors should maximally reflect the differences among the feature dimensions while retaining inter-dimension correlations. In terms of the embedding space, this means the base vectors should be uniformly distributed over the hypersphere globally. This implies that the mean of the base vectors should approximate the origin. Therefore, we designed a separate loss for $DE(\cdot)$ as follows:

$$\mathcal{L}_{DE}(T) = ||\frac{1}{d}\sum_{i=1}^{d} t_i - \mathbf{0}||_2^2, \tag{8}$$

where $d$ represents the number of dimensions, $T$ denotes the basis transformation matrix. Eq. 8 ensures the global uniformity of the basis transformation vectors, while the local similarity or dissimilarity is controlled by the dimension's inherent information and the overall model's loss.

**Graph Encoder GE$(\cdot)$.** Following numerous existing works [5, 32, 33], we implement our graph encoder using GNNs, which effectively encode the feature and structural information of graphs. We use $H = XT$ as the node feature embedding input to $GE(\cdot)$. This approach standardizes any shape of node features to the same dimension, making $GE(\cdot)$ generally applicable across various datasets.

**Relative Relationship Optimization Task $\mathcal{L}_{RT\text{-}FUG}$.** Based on the analysis of $\mathcal{L}_{RT}$ in Section 3 , the $\mathcal{L}_{RT\text{-}FUG}$ loss should be divided into positive and negative parts, $\mathcal{L}_{RT\text{-}FUG^+}$ and $\mathcal{L}_{RT\text{-}FUG^-}$ . For negative part, although the InfoNCE loss has demonstrated strong performance in prior studies [8, 5], it faces challenges with false negative sampling [34, 35] and high computational complexity (which is $O(n^2)$, see Appendix B.1). Moreover, existing studies have shown that InfoNCE-based loss functions face

conflicts in graph data, making the model gradient unable to descend [36]. Thus, we propose a new type of loss based on the analysis in $\mathcal{L}_{RT}$. Ideally, representations should be able to be divided into multiple clusters based on their intrinsic semantics, with nodes within the same cluster close together (positive pairs) and nodes between clusters far apart (negative pairs). Without prior knowledge, accurately identifying negative pairs is difficult and prone to sampling bias. Therefore, we opt not to explicitly determine negative pairs but instead constrain the overall representations, as:

$$\mathcal{L}_{\text{RT-FUG}^-} = ||\frac{1}{n} \sum_{i=1}^{n} \text{Norm}(z_i) - \mathbf{0}||_2^2, \tag{9}$$

where $z_i$ represents the embedding of node $v_i$. Similar to Eq. 8, we first constrain the representations to a unit sphere and then force the mean of the representations to be close to the origin. From Eq. 9, we reduce the complexity of the negative sampling in InfoNCE loss which is $O(n^2)$ to $O(n)$.

For positive sampling, inspired by existing work [37, 31, 30] and based on the natural homophily assumption in graphs, which means connected nodes have similar semantics, we consider node neighbors as positive samples. As Eq. 8 and 9 both use the $l2$ distance, for loss balance, we also adopt $l2$ loss here:

$$\mathcal{L}_{\text{RT-FUG}^+} = \frac{1}{|\mathcal{E}|} \sum_{(v_i, v_j) \in \mathcal{E}} ||z_i - z_j||_2^2, \tag{10}$$

where $\mathcal{E}$ represents the set of edges. Note that Eq 9 focuses on encoding global differences, while Eq. 10 focuses on encoding local homogeneity, complementing each other without conflict. The relationship between of $\mathcal{L}_{RT}$ and $\mathcal{L}_{\text{RT-FUG}}$ is in Appendix A.3. The final loss is defined as:

$$\mathcal{L}_{\text{FUG}} = \lambda_1 \mathcal{L}_{\text{RT-FUG}^-} + \lambda_2 \mathcal{L}_{\text{RT-FUG}^+} + \lambda_3 \mathcal{L}_{\text{DE}}, \tag{11}$$

where, $\lambda_1, \lambda_2, \lambda_3$ are hyper-parameters which adjust the weight of the three losses. Compared to the classical InfoNCE contrastive loss, $\mathcal{L}_{\text{FUG}}$ better encodes the relative relationships between nodes and offers a lower computational complexity of $O(n + e + d)$ (Please see Appendix B.1 for details). Notably, although the final loss involves three hyper-parameters, in fact, FUG has only one additional hyper-parameter compared to classical GCLs (or even fewer, as FUG does not require tuning multiple augmentation-related parameters). Details can be found in Appendix E.3.

Table 2: Cross-domain Node Classification. The performances of OFA and GraphControl are reported from [17, 18]. FUG-C (Cora) means FUG-C (Cora) means FUG-C model pre-trained on Cora dataset.

| Method | Cora | CiteSeer | PubMed | Photo | Computers | CS | Physics |
|---|---|---|---|---|---|---|---|
| OFA [17] | 75.90 ± 1.26 | - | 78.25 ± 0.71 | - | - | - | - |
| GraphControl [18] | - | - | - | 89.66 ± 0.56 | - | - | 94.31 ± 0.12 |
| FUG-C (Cora) | 82.35 ± 2.60 | 67.13 ± 2.80 | 84.58 ± 0.57 | 92.78 ± 1.23 | 86.27 ± 1.60 | **92.78 ± 0.66** | 95.41 ± 0.42 |
| FUG-C (CiteSeer) | 82.28 ± 2.67 | 68.53 ± 2.43 | 84.57 ± 0.35 | 92.89 ± 1.06 | 85.76 ± 1.59 | 92.62 ± 0.80 | 95.41 ± 0.41 |
| FUG-C (PubMed) | 83.31 ± 2.72 | **69.31 ± 1.49** | **84.80 ± 0.74** | 92.82 ± 1.04 | 87.70 ± 1.18 | 92.65 ± 0.72 | 95.20 ± 0.39 |
| FUG-C (Photo) | **83.35 ± 1.94** | 68.92 ± 2.05 | 84.34 ± 0.41 | 92.93 ± 1.14 | 88.01 ± 1.38 | 92.44 ± 0.69 | 95.25 ± 0.41 |
| FUG-C (Computers) | 83.31 ± 2.25 | 69.01 ± 1.79 | 84.37 ± 0.47 | **93.11 ± 1.33** | **88.23 ± 1.31** | 92.50 ± 0.65 | **95.58 ± 0.40** |
| FUG-C (CS) | 81.73 ± 2.34 | 66.32 ± 2.64 | 84.44 ± 0.50 | 91.96 ± 1.11 | 83.28 ± 1.55 | 92.54 ± 0.78 | 95.34 ± 0.37 |
| FUG-C (Physics) | 80.85 ± 2.15 | 63.32 ± 2.28 | 84.26 ± 0.61 | 84.59 ± 1.80 | 73.43 ± 2.08 | 92.13 ± 0.66 | 95.35 ± 0.42 |
| FUG (Rebuild) | 84.45 ± 2.45 | 72.43 ± 2.92 | 85.47 ± 1.13 | 93.07 ± 0.82 | 88.42 ± 0.98 | 92.89 ± 0.45 | 95.45 ± 0.27 |

## 5 Experiments

### 5.1 Experiment Settings

To comprehensively evaluate the effectiveness of FUG, we selected commonly used graph datasets with diverse node feature shapes, including Cora, CiteSeer, PubMed [11, 12], Photo, Computers [38], CS, and Physics [39]. Detailed information about these datasets can be found in Appendix E.1. We primarily compared four types of methods: supervised GNN model (GCN [1]), classic machine learning algorithms (PCA[16], DeepWalk [40]), advanced self-supervised contrastive models (GRACE [5], BGRL [32], GBT [41] DGI [33], GCA [42], ProGCL [34]), and recent graph universal pre-training models (OFA [17], GraphControl [18]). In both self-supervised model-rebuilding tests and cross-domain transfer tests, FUG is initially trained in a self-supervised manner. The trained

model is then frozen, and node representations are generated on the test datasets. These representations are subsequently used as input for downstream predictors. For the node classification experiments, to simulate a few-shot scenario, we split the dataset into train, valid, and test sets by 10%/10%/80%. We train a simple downstream $l2$ classifier [35] using only 10% labels. Detailed experimental settings can be found in Appendix E.3.

Table 3: Intra-domain Model Re-building Node Classification. Data without standard deviation are reported from previous works [5, 42, 32]. Aside from GCN, the best results are highlighted in bold, and the second-best results are underlined. OOM indicates Out-Of-Memory on an RTX 3090 GPU.

| Method | Cora | CiteSeer | PubMed | Photo | Computers | CS | Physics |
|---|---|---|---|---|---|---|---|
| Raw features | 57.90 ± 3.90 | 57.60 ± 2.85 | 79.55 ± 0.98 | 80.99 ± 1.65 | 75.59 ± 1.69 | 89.92 ± 0.95 | 93.18 ± 0.45 |
| PCA | 56.36 ± 4.14 | 48.29 ± 3.18 | 82.80 ± 0.91 | 74.92 ± 1.82 | 71.26 ± 1.34 | 87.12 ± 0.73 | 92.48 ± 0.47 |
| DeepWalk [40] | 75.70 ± 0.00 | 50.50 ± 0.00 | 80.50 ± 0.00 | 89.44 ± 0.00 | 85.68 ± 0.00 | 84.61 ± 0.00 | 91.77 ± 0.00 |
| DeepWalk+Features | 73.10 ± 0.00 | 47.60 ± 0.00 | 83.70 ± 0.00 | 90.05 ± 0.00 | 86.28 ± 0.00 | 87.70 ± 0.00 | 94.90 ± 0.00 |
| GRACE [5] | 83.20 ± 1.87 | 70.99 ± 2.29 | 85.46 ± 0.54 | 91.93 ± 0.83 | 85.36 ± 0.82 | 91.84 ± 0.37 | OOM |
| BGRL [32] | 81.57 ± 2.07 | 70.10 ± 2.04 | 83.67 ± 0.84 | 92.34 ± 0.73 | 86.51 ± 1.53 | 92.12 ± 0.63 | 95.42 ± 0.41 |
| GBT [41] | 82.45 ± 1.97 | 70.16 ± 2.32 | 85.69 ± 1.22 | 92.44 ± 0.48 | 87.46 ± 0.82 | 92.39 ± 0.78 | 95.07 ± 0.23 |
| DGI [33] | 83.24 ± 2.12 | 71.23 ± 2.37 | 84.62 ± 0.83 | 92.32 ± 0.49 | 86.12 ± 0.73 | 92.47 ± 0.60 | 94.47 ± 0.50 |
| GCA [42] | 82.83 ± 2.29 | 72.06 ± 1.91 | 85.69 ± 0.68 | 92.63 ± 1.12 | 87.78 ± 0.78 | 92.69 ± 0.49 | OOM |
| ProGCL [34] | 82.65 ± 1.74 | 72.85 ± 2.17 | OOM | 92.87 ± 0.44 | OOM | OOM | OOM |
| FUG (Rebuild) | 84.45 ± 2.45 | 72.43 ± 2.92 | 85.47 ± 1.13 | 93.07 ± 0.82 | 88.42 ± 0.98 | 92.89 ± 0.45 | 95.45 ± 0.27 |
| GCN (Supervised) | 82.80 ± 0.00 | 72.00 ± 0.00 | 84.80 ± 0.00 | 92.42 ± 0.00 | 86.51 ± 0.00 | 93.03 ± 0.00 | 95.65 ± 0.00 |

Table 4: Ablation study of the loss functions of FUG.

| | Cora | CiteSeer | PubMed | Photo | Computers | CS | Physics |
|---|---|---|---|---|---|---|---|
| FUG w/o $\mathcal{L}_{DE}$ | 81.32 ± 2.56 | 71.74 ± 2.88 | 84.68 ± 1.19 | 57.37 ± 3.38 | 50.55 ± 1.69 | 91.25 ± 0.75 | 93.77 ± 0.40 |
| FUG w/o $\mathcal{L}_{RT\text{-}FUG^-}$ | 83.31 ± 2.85 | 68.44 ± 2.63 | 85.18 ± 1.01 | 92.80 ± 0.71 | 88.29 ± 1.09 | 92.71 ± 0.53 | 95.18 ± 0.29 |
| FUG w/o $\mathcal{L}_{RT\text{-}FUG^+}$ | 84.30 ± 2.88 | 70.48 ± 2.47 | 85.24 ± 1.05 | 92.89 ± 0.82 | 88.27 ± 0.86 | 92.74 ± 0.47 | 95.29 ± 0.19 |
| FUG | 84.45 ± 2.45 | 72.43 ± 2.92 | 85.47 ± 1.13 | 93.07 ± 0.82 | 88.42 ± 0.98 | 92.89 ± 0.45 | 95.45 ± 0.27 |

## 5.2 Evaluating on Cross-domain Learning

We conduct experiments to test FUG's performance in cross-domain learning, as shown in Table 2. FUG-C represents a fixed FUG model where all hyperparameters remain constant during pre-training. This presents a more challenging scenario, as it means we cannot adjust parameters based on changes in the training datasets. In Table 2, we report the optimal accuracy of the state-of-the-art graph universal pre-training models, OFA and GraphControl, which are capable of training across datasets with different node features. These models have access to richer data during pre-training than FUG, which is not fair to FUG. However, as Table 2 shows, FUG's overall performance in cross-domain learning significantly surpasses both models. This is because OFA and GraphControl employ data preprocessing methods that significantly degrade the information in the node features (OFA uses LLM to reshape node features to the same dimension, and GraphControl replaces original node features with feature similarity). This validates our views made in Section 1 and Appendix D. It also demonstrates that FUG's dimension encoder can almost losslessly adapt to graph data with various node feature shapes.

Additionally, we derived some interesting findings from Table 2: 1). Counterintuitively, FUG-C's performances are not necessarily the best when pre-trained and tested on the same dataset. This is mainly because our loss functions focus on the relative semantic relationships between nodes, allowing the encoder to learn the knowledge of relative relationships. This leads to: **a)** This knowledge is generalizable across different datasets, so even though the datasets are from different domains, this knowledge remains the same. In other words, relative semantic relationship knowledge has weak cross-domain limitations, as stated in Section 3. **b)** The learning of relative semantic knowledge primarily relies on the distributional differences between nodes and between different feature dimensions, rather than on the absolute semantics contained in the data. Therefore, it is common in Table 2 to see that FUG-C pre-trained on other datasets performs better than when pre-trained and tested on the same dataset. 2). The performances of cross-domain trained FUG-C are close to that of FUG which is trained and tested on the same datasets with tuned hyper-parameters. In

fact, on the Photo dataset, FUG-C (Computers) outperforms FUG. This demonstrates FUG's strong cross-domain learning capability and the success of our proposed dimension encoding and relative relationship loss. It proves FUG's robust applicability to various graph data.

## 5.3 Evaluating on In-domain Learning

In Table 3, we designed node classification experiments to evaluate FUG in intra-domain self-supervised scenarios. We also report the computational overhead of FUG when the hidden layer is fixed to 256 as shown in Table 5, and the complete results are in Appendix B.2. From Table 3 and 5, we can observe that FUG demonstrates competitive performance. In Table 3, we observed that: 1). FUG outperforms GRACE across all datasets, reaffirming that the learnable DE($\cdot$) encoder in FUG can almost losslessly standardize node fea-

Table 5: Comparison of computational cost

|  | GRACE | BGRL | GBT | FUG |
|---|---|---|---|---|
| $X^-$ | ✓ | ✗ | ✗ | ✓ |
| CS Time | 0.2169 | 0.0462 | 0.0593 | 0.0526 |
| CS VRAM | 11,960 | 4,482 | 3,324 | 2,482 |
| Physics Time | OOM | 0.1084 | 0.1320 | 0.1047 |
| Physics VRAM | OOM | 10,278 | 8,002 | 5,190 |

tures to the same shape. This also supports our views in Section 3 that both positive and negative sampling in the InfoNCE loss inherently constrain the relative relationships between nodes. Furthermore, it demonstrates that our loss function maintains excellent performance while reducing computational overhead. 2). FUG significantly surpasses PCA on all datasets. Although PCA, FUG, and GRACE follow the same framework, PCA is limited by its inability to encode graph structure and the relative semantics between data points, leading to poor performance. This validates our previous views. FUG simulates PCA's encoding process through graph contrastive learning, thereby addressing PCA's shortcomings and significantly improving performance. 3). FUG generally outperforms the negative-sample-free GBT and BGRL, while maintaining comparable time costs as shown in Table 5. This indicates that considering negative relative relationships between data points is effective, and FUG successfully reduces the time and space overhead associated with these negative considerations to match that of methods that do not consider negative samples. Additionally, we compare FUG with latest advanced in-domain self-supervised methods [43, 44, 45] in Appendix E.4.

## 5.4 Model Analysis

To validate the effectiveness of each component in FUG, we conducted ablation studies and parameter analysis, as shown in Table 4 and Fig. 3. From Table 4, it is evident that FUG outperforms other models without part of loss functions across all datasets, demonstrating the effectiveness of the three proposed loss functions. Furthermore, we observed that FUG w/o

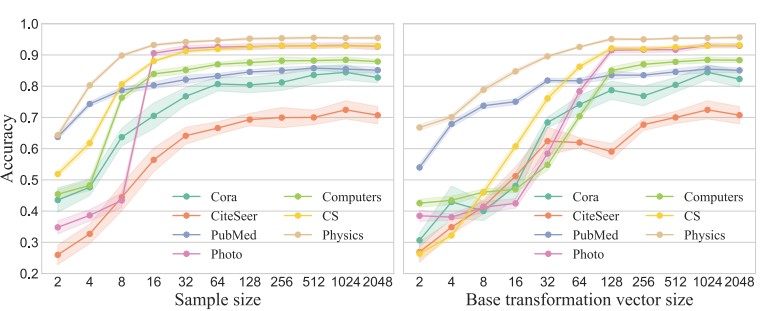

Figure 3: Hyper-parameter analysis of FUG.

$\mathcal{L}_{\text{RT-FUG}+}$ generally achieves the second-best performance. This is primarily due to $\mathcal{L}_{\text{RT-FUG}-}$ constraining global uniformity without focusing on the distance between local node pairs, preventing the model from over-encoding differences between nodes. Additionally, FUG w/o $\mathcal{L}_{\text{DE}}$ generally exhibits the worst results. The main reason is that, $L_{RT-FUG+}$ and $L_{RT-FUG-}$ are effective when based on $L_{DE}$, especially for datasets with high quality features. As described in Section 5.4, when $L_{DE}$ is absent, DE is not sufficiently trained, which leads to DE embedding $X$ almost randomly to generate $H$, which is an important input for GE. Furthermore, without the guidance of $L_{DE}$, GE only needs to extract relationships in $H$ to reduce the loss, ignoring whether $H$ contains sufficient information from $X$. Therefore, in FUG w/o $L_{DE}$, the $H$ input to GE is of poor quality, resulting in worst performances on some datasets. From Fig. 3, we can observe that as the number of sampled nodes and the dimensionality of the basis transformation vectors increase, the overall model performance

improves until it stabilizes at around 1024. This indicates that a limited number of sampled nodes can effectively represent the overall data distribution. Moreover, once the dimensionality of the basis transformation vectors reaches a certain threshold, it can embed dimensional information nearly losslessly, thereby unifying data features to the same shape. This further validates the design of the dimensional encoder.

## 6    Discussions and Future Works

In this paper, we explore the connection between contrastive learning and PCA, and propose the FUG strategy, which includes specific dimension encoding, inter-dimensional encoding, and relative semantic relationship loss. The models instantiated under this framework achieve near-lossless unification of node attributes with different shapes without data preprocessing, making them applicable to graph data and demonstrating strong cross-domain learning capabilities. We instantiated a FUG model and validated its effectiveness through extensive experiments. Additionally, our theoretical analysis shows that the complexity of the GCL loss involving negative samples is reduced from $O(n^2)$ to $O(n)$, further lowering FUG's training costs.

In future works, the FUG strategy can support research on graph prompt tuning, universal graph models, and general large models based on graph networks, enabling graph models to train directly on various data. Moreover, upon acceptance of this paper, we will release a pre-trained FUG model on existing seven datasets, allowing for direct use of FUG-generated graph representations as priors for further extensions. While the FUG strategy is general, the instantiated model has some limitations, such as only considering homophilic graphs in positive sampling, which may perform poorly on heterophilic graphs, and not constraining absolute semantic information in the loss function. Future work can further refine the universal graph contrastive pre-training model based on the FUG strategy.

## Acknowledgement

This work was supported by the National Natural Science Foundation of China [grants numbers 62422210, 62276187, 92370111, 62372323, 62472311] and the Ant Group Research Fund (2023061517131).

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

# A  Proofs

## A.1  Proof for Theorem 3.1

The proof of Theorem 3.1 is divided into two parts: framework equivalence and objective equivalence. We first present the framework equivalence part, where PCA can be decomposed into the $\text{Fea}(\cdot)$ and $\text{Dim}(\cdot)$ components.

*Proof:*

Given an input data $X$ with a standardized matrix $\widetilde{X} = X - Mean(X)$, PCA is as follows.

$$PCA(X) = \widetilde{X}P, P = Norm[Concat(c_i|c_i \in C, \lambda_c \geq \lambda^{(k)})], (\Lambda, C) = \delta[Cov(\widetilde{X})], \quad (12)$$

where $(\Lambda, C)$ represents the eigenvalues and eigenvectors matrix, $Norm(\cdot)$ represents normalization, $Concat(\cdot)$ represents concatenation, $\lambda^{(k)}$ represents the k-th largest eigenvalue, $Cov(\cdot)$ represents covariance computation, and $\delta(\cdot)$ represents eigenvalue and eigenvector computation. This can be decomposed into the forms of $\text{Dim}(\cdot)$ and $\text{Fea}(\cdot)$ as follows:

$$
\begin{aligned}
DimEmb = Encode_1(\widetilde{X}) = \widetilde{X}^T W_1, W_1 = \widetilde{X}, \\
FeaEmb = Encode_2(\widetilde{X}) = PCA(X) = \widetilde{X}W_2, W_2 = P = \varphi(DimEmb),
\end{aligned}
\quad (13)
$$

where $W_1$ is a special parameter matrix, $W_1 = \widetilde{X}$, and $\varphi(\cdot)$ represents the operation to solve $W_2$ as:

$$W_2 = P = \varphi(DimEmb) = Norm[Concat\{c_i|c_i \in C, \lambda_c \geq \lambda^{(k)}\}], (\Lambda, C) = \delta(DimEmb). \quad (14)$$

From the above, $Encode_1(\cdot)$ and $Encode_2(\cdot)$ satisfy linear or nonlinear mappings. So, we derive:

$$PCA(X) = Fea(\widetilde{X}) = \widetilde{X}W_2 = \widetilde{X}\delta(Dim(\widetilde{X})) = Fea(\widetilde{X}, Dim(\widetilde{X})). \quad (15)$$

Thus, PCA can be rigorously decomposed into $\text{Dim}(\cdot)$ and $\text{Fea}(\cdot)$ as described in Theorem 3.1.

$$\square$$

Then, we prove that $\text{Minimize}\mathcal{L}_{PCA} \propto \text{Minimize}\mathcal{L}_{RT}$.

*Proof:*

We start with the loss function of PCA, which maximizes the variance:

$$
\begin{aligned}
\text{Minimize } [\mathcal{L}_{PCA}] &= \text{Minimize } [-\mathbb{E}[||PCA(x_i) - mean(PCA(X))||_2^2]] \\
&= \text{Minimize } [-\mathbb{E}[||z_i - mean(Z)||_2^2]],
\end{aligned}
\quad (16)
$$

where $Z$ denotes the PCA embeddings. To obtain a form corresponding to Eq. 3, we consider an arbitrary pair of data points $z_i$ and $z_j$, as shown below::

$$
\begin{aligned}
\text{Minimize } [\mathcal{L}_{PCA}(z_i, z_j)] &= \text{Minimize } [-||z_i - \frac{z_1 + z_2 + \cdots + z_i + \cdots + z_j + \cdots + z_n}{n}||_2^2] \\
&= \text{Minimize } [-||z_i - \frac{z_i + z_j + c}{n}||_2^2] \\
&= \text{Minimize } [-||\frac{n-1}{n}z_i - \frac{1}{n}z_j + c||_2^2]
\end{aligned}
\quad (17)
$$

Where $c$ represents the sum of the representations of all other nodes. From Eq. 2, $Z = \widetilde{X}P$ and $Mean(\widetilde{X}) = 0$. So $Mean(\widetilde{X}P) = Z = 0$. Therefore, we approximate c to 0 here. The following

equation can be derived:

$$
\begin{aligned}
\text{Minimize } \mathcal{L}_{PCA}(z_i, z_j) &= \text{Minimize } [-||\frac{n-1}{n}z_i - \frac{1}{n}z_j + c||_2^2] \\
&\approx \text{Minimize } [-||\frac{n-1}{n}z_i - \frac{1}{n}z_j||_2^2] \\
&= \text{Minimize } [-(\frac{(n-1)^2}{n^2}||z_i||_2^2 + \frac{1}{n^2}||z_j||_2^2 - \frac{2n-2}{n^2}z_i z_j)] \\
&= \text{Minimize } [-(\frac{(n-1)^2}{n^2}m^2 + \frac{1}{n^2}m^2 - \frac{2n-2}{n^2}z_i z_j)] \\
&= \text{Minimize } [-(\frac{(n-1)^2}{n^2}m^2 + \frac{1}{n^2}m^2 - \frac{2n-2}{n^2}m^2\frac{z_i z_j}{||z_i||_2||z_j||_2})] \\
&= \text{Minimize } [-(\frac{(n-1)^2}{n^2}m^2 + \frac{1}{n^2}m^2 - \frac{2n-2}{n^2}m^2\text{Sim}(z_i, z_j))] \\
&= -(\frac{(n-1)^2}{n^2}m^2 + \frac{1}{n^2}m^2) + \frac{2n-2}{n^2}m^2(\text{Minimize } [\text{Sim}(z_i, z_j)]) \\
&\propto \text{Minimize } [\text{Sim}(z_i, z_j)]
\end{aligned}
\tag{18}
$$

here we use the assumption that all representations have the same length $m$. When $X^+ =$, Eq. 3 can have the following form:

$$
\begin{aligned}
\text{Minimize } [\mathcal{L}_{RT}] &\propto \text{Minimize } [\mathbb{E}_{(x_i,x_j)\in X^-}[\text{Sim}(z_i, z_j)] - \mathbb{E}_{(x_i,x_j)\in\emptyset}[\text{Sim}(z_i, z_j)]] \\
&= \text{Minimize } [\mathbb{E}_{(x_i,x_j)\in X^-}[\text{Sim}(z_i, z_j)]] \\
&\propto \text{Minimize } [\mathcal{L}_{PCA}]
\end{aligned}
\tag{19}
$$

$\square$

**Assumption Analysis**. In Theorem 3.1, we assume that all representations have the same length $m$. This assumption is reasonable because in many deep learning scenarios, normalizing is a common practice, specifically normalizing the length of the embeddings to 1.

### A.2 Proofs for Theorem3.2 and Corollary 3.2.1

Similar to Appendix A.1, the instantiation part of the theorem has already been clearly explained in the main text. We consider $\text{Dim}(X) = X$ as a special form of dimensional encoding instantiation, given that CL essentially lacks a dimensional encoding component. We view the neural network in CL as the instantiation of $\text{Fea}(\cdot)$. Here, we prove that Minimize $[\mathcal{L}_{CL}] \propto$ Minimiaze $[\mathcal{L}_{RT}]$.

*Proof:* First, we define a set $X_A^+$ representing the positive pairs in contrastive learning. $(x_i, x_j) \in X_A^+$ means that $x_i$ and $x_j$ are two augmented data points derived from the same original data point. Then we have:

$$
\mathcal{L}_{\text{InfoNCE}} = -\log\left(\frac{\sum_{(z_i,z_j)\in X_A^+} e^{\text{Sim}(z_i,z_j)/\tau}}{\sum_{(z_i,z_j)\in X_A^+} e^{\text{Sim}(z_i,z_j)/\tau} + \sum_{(z_i,z_j)\in X^-} e^{\text{Sim}(z_i,z_j)/\tau}}\right)
\tag{20}
$$

where $z_i$ means the embedding of data point $x_i$. Considering a positive pair $(x_a, x_a^+) \in X_A^+$, we can obtain:

$$
\begin{aligned}
\text{Minimize } [\mathcal{L}_{\text{InfoNCE}}(x_a, x_a^+)] &= \text{Minimize } [-\log\left(\frac{e^{\text{Sim}(z_a,z_a^+)/\tau} + c_p}{e^{\text{Sim}(z_a,z_a^+)/\tau} + c_p + c_n}\right)] \\
&= \text{Minimize } [-\log\left(1 - \frac{c_n}{e^{\text{Sim}(z_a,z_a^+)/\tau} + c_p + c_n}\right)] \\
&= -\log\left(1 - \frac{c_n}{\text{Maximize } [e^{\text{Sim}(z_a,z_a^+)/\tau}] + c_p + c_n}\right) \\
&= -\log\left(1 - \frac{c_n}{e^{\text{Maximize } [\text{Sim}(z_a,z_a^+)]/\tau} + c_p + c_n}\right) \\
&\propto \text{Maximize } [\text{Sim}(z_a, z_a^+)]
\end{aligned}
\tag{21}
$$

where $c_p, c_n$ are constants Considering a negative pair $(x_a, x_a^-) \in X^-$, we can obtain:

$$\text{Minimize } [\mathcal{L}_{\text{InfoNCE}}(x_a, x_a^-)] = \text{Minimize } \left[ -\log \left( \frac{c_p}{e^{\text{Sim}(z_a, z_a^-)/\tau} + c_p + c_n} \right) \right]$$

$$= -\log \left( \frac{c_p}{\text{Minimize } [e^{\text{Sim}(z_a, z_a^-)/\tau}] + c_p + c_n} \right) \qquad (22)$$

$$= -\log \left( \frac{c_p}{e^{\text{Minimize } [\text{Sim}(z_a, z_a^-)]/\tau} + c_p + c_n} \right)$$

$$\propto \text{Minimize } [\text{Sim}(z_a, z_a^-)]$$

So, we have:

$$\text{Minimize } [\mathcal{L}_{\text{InfoNCE}}] \propto \text{Minimize } \left[ \sum_{(x_i, x_j) \in X^-} \text{Sim}(z_i, z_j) \right] \& \text{Maximize } \left[ \sum_{(x_i, x_j) \in X_A^+} \text{Sim}(z_i, z_j) \right]$$

$$\propto \text{Minimize } \left[ \sum_{(x_i, x_j) \in X^-} \text{Sim}(z_i, z_j) - \sum_{(x_i, x_j) \in X_A^+} \text{Sim}(z_i, z_j) \right]$$

(23)

Here we get the conclusion that minimizing the contrastive InfoNCE loss is maximizing the similarity between positive pairs and minimizing the similarity between negative pairs. Based on our assumption, data augmentation does not alter the essential semantics of the data. We define $H$ as the data stripped of all irrelevant and redundant information by data augmentation, containing only the essential semantics. For a data point $x_a$, maximizing $\sum_{(x_a, x_j) \in X_A^+} \text{Sim}(x_a, x_j)$ is equivalent to maximizing $\sum_{(x_a, x_j) \in X_A^+} \text{Sim}(x_a, h_a)$. For downstream tasks, data of the same category embodies the same essential semantics. Therefore, for a set $\mathcal{D}$ of data points have similar essential semantics, their $h$ values are identical. For any two data points $x_a, x_b \in \mathcal{D}_{\|}$, we have:

$$\text{Minimize } [\mathcal{L}_{\text{InfoNCE}}] \propto \text{Minimize } \left[ \sum_{(x_i, x_j) \in X^-} \text{Sim}(z_i, z_j) - \sum_{k=1}^{n_k} \sum_{x_i \in \mathcal{D}_{\|}} \text{Sim}(z_i, z_i^+) \right]$$

$$= \text{Minimize } \left[ \sum_{(x_i, x_j) \in X^-} \text{Sim}(z_i, z_j) - \sum_{k=1}^{n_k} \sum_{x_i \in \mathcal{D}_{\|}} \text{Sim}(z_i, h_k) \right]$$

$$= \text{Minimize } \left[ \sum_{(x_i, x_j) \in X^-} \text{Sim}(z_i, z_j) - \sum_{k=1}^{n_k} \sum_{x_i, x_j \in \mathcal{D}_{\|}} \text{Sim}(z_i, z_j) \right] \qquad (24)$$

$$= \text{Minimize } \left[ \sum_{(x_i, x_j) \in X^-} \text{Sim}(z_i, z_j) - \sum_{x_i, x_j \in X^+} \text{Sim}(z_i, z_j) \right]$$

$$\propto \text{Minimize } [\mathcal{L}_{RT}]$$

Comparing equations 19 and 24, we can conclude that the PCA optimization objective is, in fact, a special case of the contrastive loss when $X^+ = \emptyset$. $\qquad \square$

**Assumption Analysis**. We assume that data augmentation does not alter the essential semantics of the data, which is an idealized condition. Many works in contrastive learning aim to achieve this through augmentation. Therefore, this assumption is reasonable. Additionally, this assumption is made for the sake of clarity. Under this assumption, the set $\mathcal{D}$ is correctly defined, meaning all data within the set share the same essential semantics. If the assumption does not hold, the essential semantics within the set $\mathcal{D}$ may not be identical across all nodes. However, this does not affect the correctness of the proof, as the set $\mathcal{D}$ still exists.

## A.3 The Relationship Between the RT-FUG Loss and RT Loss

In this section, we discuss the relationship between $\mathcal{L}_{\text{RT-FUG}}$ and $\mathcal{L}_{RT}$.

**Theorem A.1** *Assuming the representations have the same length $m$, then:*

$$\text{Minimize } [\mathcal{L}_{\text{RT-FUG}}] \propto \text{Minimize } [\mathcal{L}_{RT}] \qquad (25)$$

*Proof:*

The $\mathcal{L}_{\text{RT-FUG}}$ consists of both positive and negative parts, which we discuss separately. For $\mathcal{L}_{\text{RT-FUG}^+}$, we have:

$$
\begin{aligned}
\text{Minimize } [\mathcal{L}_{\text{RT-FUG}^+}] &= \text{Minimize } [\sum_{(v_i,v_j)\in X^+} ||z_i - z_j||_2^2] \\
&= \text{Minimize } [\sum_{(v_i,v_j)\in X^+} (z_i^2 + z_j^2 - 2z_iz_j)] \\
&= \text{Minimize } [\sum_{(v_i,v_j)\in X^+} (2m^2 - 2m^2\frac{z_i,z_j}{||z_i||_2||z_j||_2})] \\
&= \text{Minimize } [\sum_{(v_i,v_j)\in X^+} (2m^2 - 2m^2\text{Sim}(z_i,z_j))] \\
&= 2n_{X^+}m^2 - 2m^2\,\text{Maximize } [\sum_{(v_i,v_j)\in X^+} (\text{Sim}(z_i,z_j)])] \\
&\propto \text{Minimize } [-\sum_{(v_i,v_j)\in X^+} \text{Sim}(z_i,z_j)]
\end{aligned}
\tag{26}
$$

For $\mathcal{L}_{\text{RT-FUG}^-}$ take any two nodes $v_i$, $v_j$ then we have:

$$
\begin{aligned}
\text{Minimize } [\mathcal{L}_{\text{RT-FUG}^-}(v_i,v_j)] &= \text{Minimize } [||\frac{\frac{z_1}{m} + \frac{z_2}{m} + \cdots + \frac{z_i}{m} + \cdots + \frac{z_j}{m} + \cdots + \frac{z_n}{m}}{n} - \mathbf{0}||_2^2] \\
&= \text{Minimize } [||\frac{1}{n}(\frac{z_i}{m} + \frac{z_j}{m} + c) - \mathbf{0}||_2^2] \\
&= \text{Minimize } [||\frac{1}{n}(\frac{z_i}{m} + \frac{z_j}{m} + c)||_2^2]
\end{aligned}
\tag{27}
$$

here, we also use a similar trick as in Appendix A.1, treat the constant vector $c$ as $\mathbf{0}$:

$$
\begin{aligned}
\text{Minimize } [\mathcal{L}_{\text{RT-FUG}^-}(v_i,v_j)] &= \text{Minimize } [||\frac{1}{n}(\frac{z_i}{m} + \frac{z_j}{m} + c)||_2^2] \\
&\approx \text{Minimize } [||\frac{1}{n}(\frac{z_i}{m} + \frac{z_j}{m})||_2^2] \\
&= \text{Minimize } [\frac{1}{nm}||z_i + z_j||_2^2] \\
&= \text{Minimize } [\frac{1}{nm}(z_i^2 + z_j^2 + 2z_iz_j)] \\
&= \text{Minimize } [\frac{1}{nm}(2m^2 + 2z_iz_j)] \\
&= \text{Minimize } [\frac{1}{nm}(2m^2 + 2m^2\frac{z_iz_j}{||z_i||_2||z_j||_2})] \\
&= \text{Minimize } [\frac{1}{nm}(2m^2 + 2m^2\,\text{Sim}(z_i,z_j))] \\
&= \frac{1}{nm}(2m^2 + 2m^2\text{Minimize } [\text{Sim}(z_i,z_j)]) \\
&\propto \text{Minimize } [\text{Sim}(z_i,z_j)]
\end{aligned}
\tag{28}
$$

Hence, from Eq. 28 and 26, we can obtain:

$$
\begin{aligned}
\text{Minimize } [\mathcal{L}_{\text{RT-FUG}}] &= \text{Minimize } [\mathcal{L}_{\text{RT-FUG}^+} + \mathcal{L}_{\text{RT-FUG}^-}] \\
&\propto \text{Minimize } [\mathcal{L}_{\text{RT-FUG}^+}] + \text{Minimize } [\mathcal{L}_{\text{RT-FUG}^-}] \\
&\propto \text{Minimize } [\mathcal{L}_{\text{RT-FUG}^+}] + \text{Minimize } [\mathcal{L}_{\text{RT-FUG}^-}] \\
&\propto \text{Minimize } [\mathbb{E}_{(x_i,x_j)\in X^-}[\text{Sim}(z_i,z_j)] - \mathbb{E}_{(x_i,x_j)\in X^+}[\text{Sim}(z_i,z_j)]] \\
&\propto \text{Minimize } [\mathcal{L}_{RT}]
\end{aligned}
\tag{29}
$$

$\square$

Table 6: Comparison of Computational Cost (Completed)

|  | GRACE | BGRL | GBT | FUG |
|---|---|---|---|---|
| $X^-$ | ✓ | ✗ | ✗ | ✓ |
| Cora Time | 0.0106 | 0.0055 | 0.0095 | 0.0091 |
| Cora VRAM | 680 | 532 | 526 | 674 |
| CiteSeer Time | 0.0153 | 0.0075 | 0.0126 | 0.0148 |
| CiteSeer VRAM | 910 | 788 | 658 | 848 |
| PubMed Time | 0.2154 | 0.0201 | 0.0257 | 0.0165 |
| PubMed VRAM | 11,610 | 1,254 | 1,432 | 1,340 |
| Photo Time | 0.0552 | 0.0225 | 0.0270 | 0.0220 |
| Photo VRAM | 3,692 | 1,336 | 2,158 | 1,962 |
| Computers Time | 0.1397 | 0.0440 | 0.0492 | 0.0407 |
| Computers VRAM | 8,952 | 2,316 | 2,570 | 6,418 |
| CS Time | 0.2169 | 0.0462 | 0.0593 | 0.0526 |
| CS VRAM | 11,960 | 4,482 | 3,324 | 2,482 |
| Physics Time | OOM | 0.1084 | 0.1320 | 0.1047 |
| Physics VRAM | OOM | 10,278 | 8,002 | 5,190 |

# B   Analysis of Computational Costs

## B.1   Computational Complexity

**RT-FUG Loss.** As shown in Eq. 9, $\mathcal{L}_{\text{RT-FUG}}$ in the negative sampling part only requires calculating the mean of all node representations after normalization and computing the distance to the origin. In fact, since $\text{Mean}(\cdot) - \mathbf{0} = \text{Mean}(\cdot)$, we only need to perform operations directly on the mean. Thus, its computational complexity is $O(n)$. The negative sampling part of the InfoNCE loss is well-known for its high complexity, as it must compute the similarity between every pair of nodes, resulting in $O(n^2)$ complexity. Therefore, we state in the main paper that $L_{\text{RT-FUG}}$, as an optimized contrastive loss, reduces the original time complexity from $O(n^2)$ to $O(n)$.

As shownin Eq. 10, for the positive sampling part, we discard the costly data augmentation and, guided by Theorem 3.2, explicitly define neighbors as positive samples. Hence, we only need to compute the distance for the two points linked by an edge. Its time complexity is $O(e)$, where $e$ is the number of edges. Therefore, the overall complexity of the RT-FUG loss is $O(n + e)$.

**DE Loss.** Similar to $\mathcal{L}_{\text{RT-FUG}^-}$, we compute the mean of the basis transformation matrix for each dimension and its distance to the origin. Thus, its time complexity is $O(d)$.

**Overall Time Complexity.** From the above, the overall time complexity of the loss is $O(n + e + d)$. Furthermore, considering $d$ is the number of dimensions of node features, it is negligible compared to the number of nodes $n$ when the data is large enough. Additionally, even if the number of nodes is very large, the average degree of nodes is generally small. Assuming the average node degree is a fixed constant $a$, the overall time complexity becomes $O((2a + 1)n)$, which can be regarded as $O(n)$. For accuracy, we still state the complexity of $L_{\text{FUG}}$ as $O(n + e + d)$ in the main paper.

## B.2   Computational overhead experiments

In this section, we present the complete computational cost experiments reported in Table 3 of the main text, as shown in Table 6. The implementation details of the comparative methods are provided in Appendix E.2. To eliminate the influence of hidden layer size on model comparison, we fixed the hidden layer to 256 dimensions and reported the memory usage and computational cost per epoch. As observed in the table, FUG, which considers negative samples, has time and space complexity comparable to methods that only consider positive samples. Furthermore, FUG's advantages become more pronounced as the number of nodes in the dataset increases. As stated in Appendix B.1, when

the dataset is large enough, the time complexity of FUG is $O(n)$ rather than $O(n+e+d)$. Therefore, FUG performs comprehensively best on the Physics dataset. However, on smaller datasets like Cora and CiteSeer, FUG's advantage is not as evident because the feature dimensions are large relative to the number of nodes. Table 6 further demonstrates FUG's superiority in computational efficiency.

## C   Algorithm

To further illustrate how FUG works, we provide pseudocode as shown in Algorithm 1

---

**Algorithm 1** FUG Training

---

**Input:** Data $\{X^{(g)}, A^{(g)} | (X^{(g)}, A^{(g)}) \in \mathcal{G}^{(g)}, g = 1, 2, ..., g_n\}$;
  Dimensional encoder $\mathrm{DE}_\theta(\cdot)$; Graph encoder $\mathrm{GE}_\phi(\cdot)$;
  Hyper-parameters: Learning rate $\alpha$; Loss weights $\lambda_1, \lambda_2, \lambda_3$
**Output:** Trained $\mathrm{DE}_\theta(\cdot)$ and $\mathrm{GE}_\phi(\cdot)$;

 1: **for** Number of training epochs **do**
 2:   **for** $g = 1, 2, ..., g_n$ **do**
 3:    /* Generate basis transformation matrix */
 4:    $\widehat{X^{(g)}} = \mathrm{Sample}(X^{(g)})$
 5:    $T = \mathrm{Norm}(\mathrm{DE}_\theta(\widehat{X^{(g)}}^T))$
 6:    $H = X^{(g)}T$
 7:    $Z = \mathrm{GE}_\phi(H, A(g))$
 8:    /* Calculate loss */
 9:    $\mathcal{L}_{\mathrm{RT\text{-}FUG}^-} = ||\mathrm{Mean}(\mathrm{Norm}(Z))||_2^2$
 10:    $\mathcal{L}_{\mathrm{RT\text{-}FUG}^+} = \sum_{A_{i,j}!=0} ||z_i - z_j||_2^2$
 11:    $\mathcal{L}_{\mathrm{DE}} = ||\mathrm{Mean}(T)||_2^2$
 12:    $\mathcal{L}_{\mathrm{FUG}} = \lambda_1 \mathcal{L}_{\mathrm{RT\text{-}FUG}^-} + \lambda_2 \mathcal{L}_{\mathrm{RT\text{-}FUG}^+} + \lambda_3 \mathcal{L}_{\mathrm{DE}}$
 13:    /* Update parameters */
 14:    $\phi \leftarrow \phi - \alpha \Delta_\phi \mathcal{L}; \quad \theta \leftarrow \theta - \alpha \Delta_\theta \mathcal{L}$
 15:   **end for**
 16: **end for**
 17: **return** $\mathrm{DE}_\theta(\cdot), \mathrm{GE}_\phi(\cdot)$

---

## D   Related works

**Graph Contrastive Learning (GCL).** Graph Contrastive Learning is dedicated to learning graph representations in a self-supervised manner [35]. Initially inspired by the success of contrastive learning in the visual domain [8, 46, 47], approaches such as GRACE [5] were proposed. These methods generate augmented views by randomly perturbing graph data, treating nodes augmented from the same original node as positive samples, and other nodes as negative samples. They employ the InfoNCE loss [48] to maximize mutual information between positive pairs while minimizing it between negative pairs. Numerous improvements have been made within this framework. To address the computational burden of negative sampling, some studies [32, 41] have explored training methods that do not require negative sampling. BGRL [32] utilizes two distinct encoders, Online and Target, with the Online encoder learning to replicate the Target's output for positive samples. GBT [41] proposes a loss to reduce redundancy in node representations. To mitigate the semantic distortion caused by random graph data augmentations, GCA [42] assesses the importance of nodes based on degree centrality, increasing perturbations for less important nodes. HomoGCL [28] uses community detection algorithms to assess the importance of edges and node features for designing augmentation. AD-GCL [49] introduces a learnable data augmentation strategy through a minimax approach. SpCo [50] proposes a graph spectral-based augmentation strategy. Works like AFGRL [31], AFGCL [30], and NeCo [29] have introduced contrastive learning algorithms that do not use data augmentations. To resolve sampling biases in positive and negative sampling, LocalGCL [37] and NeCo [29] suggest using neighbors as positive samples. ProGCL [34] utilizes a Beta distribution fitting approach to select negative samples. While these methods have shown great success, they all use GNNs [1, 22]

as encoders, meaning the trained models are not applicable to other datasets with different forms of node features.

**Unified Graph Pre-training Models** Inspired by the recent success of unified large language models [51], numerous studies have attempted to propose unified, generic graph pre-training models for graph data. All-In-One [52] employs transferable graph prompts, allowing graph pre-training models to be applicable across various tasks, though it still cannot handle graphs with diverse features and relies on methods like PCA to standardize node feature dimensions first. OFA [17] introduces a pre-training and fine-tuning pipeline that transforms all downstream tasks into link prediction tasks, with the model itself lacking adaptability to arbitrary node features. It uses large language models to convert node features into a uniform format. Classic approaches like GCC [6] and recent methods like UniLink [15] discard original node features in favor of encoding new topology-based positional attributes, allowing application across diverse data but resulting in significant information loss. Very recently, GraphControl [18] converts distances between node features into a matrix similar to adjacency matrices, utilizing the universality of GNNs for arbitrary topologies to achieve attribute universality. However, this approach, which only encodes distances between node features, still significantly harms the semantic integrity of the original node features.

Although these methods have extended the applicability of existing GNNs, they still lead to considerable loss of node feature information during preprocessing. In contrast, our FUG retains the semantic integrity of node features while ensuring the universality of the graph pre-training model.

# E   Experimental Supplement

## E.1   Datasets

We follow many prior works [5, 42, 32, 33] and evaluated our performance on seven widely used public datasets:

- **Cora**, **CiteSeer**, and **PubMed** are the citation datasets from PlantoID [11, 12]. Nodes represent scientific papers, edges represent citation relationships, node features are bag-of-words vectors of papers, and labels represent domains of papers.

- **Photo** and **Computers** are co-purchase graphs from Amazon [38]. Nodes represent products, edges represent frequent co-purchase relationships, node features are bag-of-words vectors of product reviews, and labels represent product categories.

- **CS** and **Physics** are from the Microsoft Academic Graph [39]. The nodes represent researchers, the edges represent paper co-authorship relationships, the node features are the bag-of-words vectors of the authors' papers, and the labels represent the research fields of the authors.

The statistical information is presented in Tab. 7. We follow the exist work [5, 35], utilized the Python library "torch_geometric.dataset" to load the datasets. Bidirectional links were established for all edges. For all the data, we divided the training, validation, and test data sets with 10%, 10%, and 80%, to simulate the scenario with fewer tags downstream.

Table 7: Dataset statistics

| Name | Nodes | Edges | Features | Classes |
|---|---|---|---|---|
| Cora | 2,708 | 4,732 | 1,433 | 7 |
| CiteSeer | 3,327 | 5,429 | 3,703 | 6 |
| PubMed | 19,717 | 44,338 | 500 | 3 |
| Photo | 7,650 | 119,081 | 745 | 8 |
| Computers | 13,752 | 245,861 | 767 | 10 |
| CS | 18,333 | 81,894 | 6,805 | 15 |
| Physics | 34,493 | 247,962 | 8,415 | 5 |

## E.2 Compared Methods

In this section, we detail the implementations of the methods we compare against. For supervised GCN [1], as well as the classic Deepwalk and Deepwalk+Features [40] methods, we directly report the accuracy from [5, 42, 32] for comparison. For the "Raw features" method in Table 3, we directly use the node features from the dataset as embeddings for the downstream classifier. For the PCA [16] in Table 3, we use node features as PCA input, reduce the dimensions uniformly to 100, and then feed them into the downstream classifier. For GRACE [5], GCA [42], DGI [33], and ProGCL [34], we use the official code for implementation. To ensure a fair comparison, we replace the downstream classifiers with a unified classifier (details in Appendix E.3). For BGRL, since the official code is entirely based on "PyTorch" while other methods are based on "torch_geometric" or "DGL" libraries, we use the implementation from [31] for fair comparison. For GBT [41], since the official code is based on Docker, we use the implementation from [35] for fair comparison. For the graph pre-training models OFA [17] and GraphControl [18], due to their use of large language models or massive datasets for training, we were unable to reproduce the models due to computational constraints. Therefore, we directly report the best accuracies from their respective papers.

## E.3 Detail Settings

**Training, Embedding and Testing.** For all experiments, we first train our models in an unsupervised manner on the training dataset, and then generate representations on the test dataset which are fed into a downstream classifier. Following the prior work [35], we set up a simple $l2$ classifier, training it with only 10% of the data and validating as well as testing with the remaining 90%. Moreover, for edge-sparse datasets like Cora and CiteSeer, we adhere to the method proposed in [53] by further enhancing the similarity relations between connected nodes through topological propagation during embedding.

**Evaluation Methods.** The number of propagation layers is denoted by "num hop". All reported results are averaged over 20 runs with different splits, providing mean and variance. The mean and standard deviation are directly calculated using "torch.mean" and "torch.std", respectively.

**Installation of FUG.** For FUG in all scenarios, we use a two-layer GCN as the graph encoder. We chose Adam as the Optimizer, the learning rate is set to 0.00001, the weight decay is set to 0.00001, and PReLU is selected as the activation function. The dimension encoder is a MLP, $\text{Linear}(\text{PReLU}(\text{Linear}(\cdot)))$. The number of nodes sampled and the dimension of the basis transformation vector are both 1024. In addition, in order to reduce the fluctuation caused by random, we directly select the first 1024 nodes as the sampling nodes for all datasets. The random seed is fixed to "66666" in all scenarios. All experiments were conducted on a device equipped with an Intel 12400 CPU and an NVIDIA RTX 3090 GPU. For FUG-C in Table 2, we set the same hyper-parameters as shown in Table 8. The hyper-parameters for FUG (Rebuild) in Table 3 are as depicted in Table 9.

**The Selection of Hyper-parameters.** Although Equation 11 includes three hyper-parameters, in fact, tuning hyper-parameters of FUG is not complex. Because the FUG model has only one additional hyper-parameter compared to classical GCLs (or even fewer, as FUG does not require tuning multiple augmentation-related parameters). Specifically, (1) in Equation 11, $\lambda_1$ is fixed at 1, requiring only two hyper-parameters to be tuned, while classic GCLs require tuning the temperature parameter $\tau$. Therefore, FUG has just one more hyper-parameter to be tuned. (2) As shown in Table 9 (in the appendix), the search spaces for $\lambda_2$ and $\lambda_3$ are small, specifically [0.1, 0.3, 0.5] and [20, 50, 100, 200, 300], respectively. They exhibit clear unimodality in sensitive analysis in Figure 4, making the tuning process straightforward. (3) As shown in Table 2 and Figure 4, the model can achieve good results even without meticulous tuning, demonstrating the robustness of FUG.

Table 8: Hyper-parameters of FUG-C in cross-domain learning

|       | #Epoch | #Hidden Units | $\lambda_1$ | $\lambda_2$ | $\lambda_3$ |
|-------|--------|---------------|-------------|-------------|-------------|
| FUG-C | 500    | 1,024         | 1           | 0.5         | 200         |

Table 9: Hyper-parameters of FUG in in-domain learning

|          | #Epoch | #Hidden Units | $\lambda_1$ | $\lambda_2$ | $\lambda_3$ | Num Hop |
|----------|--------|---------------|-------------|-------------|-------------|---------|
| Cora     | 300    | 512           | 1           | 0.3         | 100         | 3       |
| CiteSeer | 500    | 512           | 1           | 0.5         | 50          | 5       |
| PubMed   | 700    | 2,048         | 1           | 0.1         | 20          | 0       |
| Photo    | 500    | 1,024         | 1           | 0.5         | 200         | 0       |
| Computers| 500    | 1,024         | 1           | 0.1         | 300         | 0       |
| CS       | 300    | 1,024         | 1           | 0.1         | 200         | 0       |
| Physics  | 300    | 1,024         | 1           | 0.1         | 200         | 0       |

### E.4 Further Comparing FUG with Recent Advanced in-domain Methods

To further showcase FUG's performance in in-domain scenarios, we compare FUG with the latest state-of-the-art algorithms, as shown in Table 10. The results for GraphACL [43] and ROSEN [44] are obtained using the same experimental settings as described above. For POT [45], a plug-in method with similar experimental settings to ours, we directly report its best performances as stated in its original paper. As shown in Table 10, even in intra-domain scenarios (which are not our primary focus), FUG demonstrates competitive performance, achieving the best results on three out of five datasets. This highlights the effectiveness of FUG and supports the claims made in our paper.

Table 10: Additional Intra-domain Model Re-building Node Classification.

| Method        | Cora            | CiteSeer        | PubMed          | Photo           | Computers       |
|---------------|-----------------|-----------------|-----------------|-----------------|-----------------|
| GraphACL [43] | 83.40 ± 2.12    | **73.37 ± 1.77**| 85.07 ± 0.62    | 92.93 ± 0.79    | **89.50 ± 0.31**|
| ROSEN [44]    | 83.35 ± 1.77    | 73.01 ± 2.01    | 85.33 ± 0.84    | 92.82 ± 0.72    | 88.87 ± 0.42    |
| POT [45]      | 80.50 ± 0.90    | 69.00 ± 0.80    | 82.80 ± 2.00    | 92.40 ± 0.30    | 89.10 ± 0.30    |
| FUG (Rebuild) | **84.45 ± 2.45**| 72.43 ± 2.92    | **85.47 ± 1.13**| **93.07 ± 0.82**| 88.42 ± 0.98    |

### E.5 Hyper-Parameters Analysis

To fully illustrate the interactions among the three losses included in FUG, we report the analysis experiments on the weight parameters $\lambda_1$, $\lambda_2$, and $\lambda_3$ across all datasets, as shown in Figure 3.

## F Limitations

Here, we describe the limitations of the proposed FUG model. The FUG model serves as a straightforward implementation to validate our proposed FUG strategy and has some imperfections. For example, the loss $L_{\text{RT-FUG+}}$ in FUG model is based on the assumption of graph homophily assumption, which leads to good performance on homophily datasets. However, it might face challenges on heterophilic datasets because neighboring nodes in such datasets often have different semantics. Additionally, the encoder used in FUG is a GCN, which also operates under the graph homogeneity assumption. Furthermore, the GCN encoder we use can only handle weighted ordinary edges and homogeneous nodes. Therefore, when dealing with heterogeneous graphs, where node types within the same dataset differ, the model still faces applicability challenges. These issues can be addressed by following the FUG strategy to construct new instantiated models, which is a direction we aim to explore in future works.

## G Border Impacts

Here, we discuss the social border impacts related to the FUG strategy and model. As a foundational study in graph pre-training models, FUG itself does not have direct Border Impacts. Furthermore, all datasets used for training the FUG model are public, eliminating privacy concerns. Given the broad applications of Graph Neural Networks (GNNs), further research on FUG could potentially advance

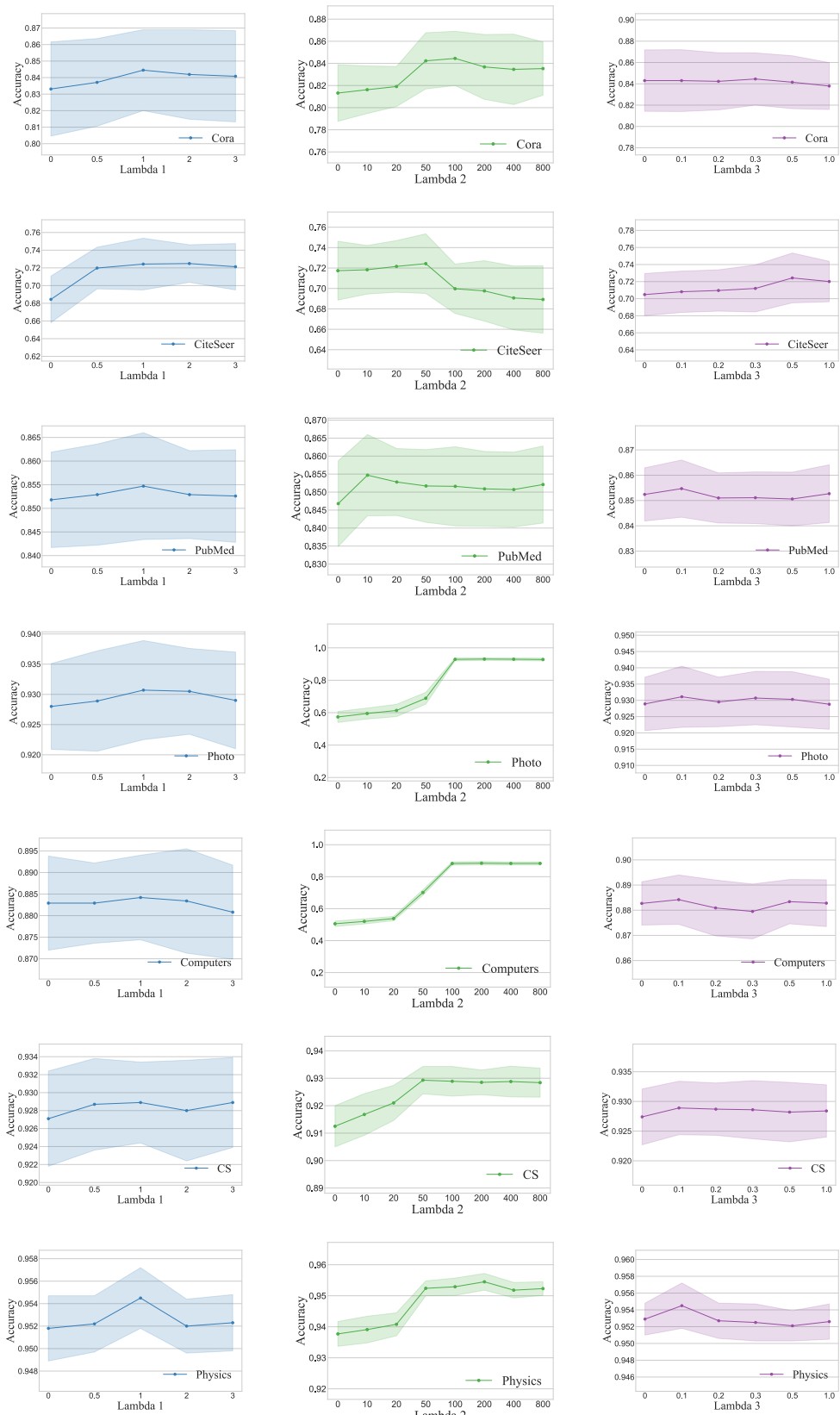

Figure 4: Parameter Analysis of $\lambda_1$, $\lambda_2$ and $\lambda_3$

multiple fields and serve as foundational research for large graph-based models. While FUG, like any neural network, might face societal risks, these risks are too remote for a preliminary theoretical study like this and should be addressed in subsequent research.

