# OpenReview forum: "FUG: Feature-Universal Graph Contrastive Pre-training for Graphs with Diverse Node Features"
_NeurIPS.cc/2024/Conference — NeurIPS 2024 poster_

### Official Review · Reviewer_C58N · 2024-07-10

**Soundness:** 3
**Presentation:** 3
**Contribution:** 3
**Rating:** 5
**Confidence:** 4

**Summary:**

In this paper, the authors design a feature-universal graph contrastive pre-training strategy for model rebuiding and data reshaping. They focus on the inherent complex characteristics in graphs, and introduce a theoretical analysis for their method. The gloabl uniformity constraint reduces the time complexity from O(n^2) to O(n).

**Strengths:**

1. This paper introduces a theoretical analysis about their designed method.

2. The authors propose a feature-universal graph contrastive pre-training strategy.

3. The designed method is clearly described.

**Weaknesses:**

1. Line 59 points out the shortcomings of the language model, but with the development of LLM, numerous LLM4graph papers have been proposed, effectively solving the problem of feature diversity.

2. The results in Table 3 demonstrate certain effectiveness of the designed method, but the methods compared are not the latest [1, 2, 3, 4]. The improvement in performance is border. The effectiveness of the design method is not convincing enough.

3. The results in Table 4 show that DE has a significant impact on model performance, while the other two parts have little impact on the model. How can the effectiveness of these two module designs be proven?

[1] Xiao, Teng, et al. "Simple and asymmetric graph contrastive learning without augmentations." Advances in Neural Information Processing Systems 36 (2024).

[2] Yu, Yue, et al. "Provable training for graph contrastive learning." Advances in Neural Information Processing Systems 36 (2024).

[3] Zhuo, Jiaming, et al. "Graph contrastive learning reimagined: Exploring universality." Proceedings of the ACM on Web Conference 2024. 2024.

[4] Bo, Deyu, et al. "Graph contrastive learning with stable and scalable spectral encoding." Advances in Neural Information Processing Systems 36 (2024).

**Questions:**

1. There is a lack of sensitivity analysis experiments for the hyperparameters in Equation 11.

2. In Equation 9, z is made to approach 0. Will this operation lose the diversity of features?

**Limitations:**

The use of Gaussian function can also achieve uniformity. Please analyze the advantages and disadvantages compared to the method designed in the paper.

---

> ### Author Rebuttal · Authors · 2024-08-06
>
> Thanks for your valuable comments. Our responses are as follows.
> # R1 for W1.
> While there have been some LLM4graphs works which seems achieve feature diversity problem, as highlighted in our paper, they encounter many challenges which is correctly our focus.
> - **Restricted Feature Input.** LLMs require node features to be in text form, whereas FUG can handle node features in various formats, including an amount of floating-point data. Moreover, arbitrarily converting graphs or nodes into text can lead to serious domain biases.
> - **Structure Neglect.** LLMs are unable to process structured data, often resulting in the separate encoding of features and structures in LLM4Graphs. In contrast, FUG can simultaneously encode features and structures.
> - **Model Complexity.** Due to their large number of parameters, LLMs struggle to be optimized by graph tasks in pre-training. FUG, being lightweight, is easy to train and applicable to various scenarios.
>
> Therefore, using LLMs to solve the feature diversity problem is often limited and sub-optimal. As shown in Table 2, OFA is a recent SOTA LLM4Graph approaches, yet its performances on the text-rich graph datasets Cora and PubMed are significantly inferior to that of FUG. On the PubMed dataset, OFA's performance is even worse than simply using the raw bag-of-words node features as input for downstream tasks.
>
> Furthermore, FUG strategy may have the potential to facilitate the development of large graph models, which is more suitable for graph data.
> # R2 for W2.
> Thank you very much for the provided the references.
>
> First, please kindly note that, the main focus of our work is to develop a graph pre-training model which is generalizable across different datasets, while the four cited methods cannot be applied to cross-domain scenarios with varying feature shapes. Besides, the results in Table 2 rather than Table 3 validates our main contribution. As shown in Table 2, FUG demonstrates significant performance advantages compared to the SOTA cross-domain graph pre-training methods, which demonstrate the effectiveness of our method.
>
> In the intra-domain scenario, follow your suggestions, we compare FUG with these works, as shown in the table below. GraphACL and ROSEN results are obtained using the same experimental settings as in our paper. POT, being a plug-in method with experimental settings similar to ours, is directly reported with its best performances from its paper. Due to the unavailability of Sp2GCL code and the considerable differences in the datasets used in their paper, we do not compare with this method.
> | |Cora|Citeseer|PubMed|Photo|Computers|
> |-|-|-|-|-|-|
> |GraphACL [1]|83.40±2.12|**73.37±1.77**|85.07±0.62|92.93±0.79|**89.50±0.31**|
> |ROSEN [3]|83.35±1.77|73.01±2.01|85.33±0.84|92.82±0.72|88.87±0.42|
> |POT [2]|80.50±0.90|69.00±0.80|82.80±2.00|92.40±0.30|89.10±0.30|
> |FUG (Ours)|**84.45±2.45**|72.43±2.92|**85.47±1.13**|**93.07±0.82**|88.42±0.98|
>
> As shown in the table above, even in intra-domain scenarios (which are not our main focus), FUG demonstrates competitive performance, achieving the best results on three out of five datasets. This proves the effectiveness of FUG and is consistent with the arguments in our paper. We will add these results and discussions in the future versions.
>
> # R3 for W3.
> In fact, all the three losses are crucial and effective, which is why FUG outperforms the other three methods across all datasets in Table 4. We explain W3 from two aspects.
>
> - In fact, these three losses are not independent; the two RT-FUG losses rely on $L_{DE}$, as it ensures that the input $H$ of GE retains sufficient information from $X$. This is evident in datasets with high-quality features like Photo and Computers, where without $L_{DE}$, the other two losses do not function properly.
> - Notably, the two RT-FUG losses have a greater impact on performance in datasets with high-quality structures, such as CiteSeer. This proves the effectiveness of the RT-FUG losses, as they guide GE to encode structures effectively with a low-quality $H$.
>
> Thus, as a pre-training model suitable for a wide range of datasets, all three losses are indispensable for FUG.
> # R4 for Q1.
> In fact, we have already conducted the sensitivity analysis experiments. But due to limited space in the main text, we included the results in Appendix F, Figure 4. We will add a guide in the main text in future versions to help readers find these results.
> # R5 for Q2.
> We apologize for any misunderstanding. In fact, in Equation 9, $z$ is not made to approach $0$. Instead, we enforce $||z||_2 = 1$. Equation 9 ensures that the **mean** of the normalized representations approaches $0$, which allows the representations to be uniformly distributed on a hyper-sphere centered at the origin with a radius of 1. This guarantees the diversity of the representations.
> # R6 for L1.
> Thank you very much for pointing out the limitation we had overlooked. In our understanding, the Gaussian function you mentioned refers to the "Gaussian potential kernel" provided in "Understanding Contrastive Representation Learning through Alignment and Uniformity on the Hypersphere". (If this is not the case, we will be grateful if you could provide new references.)
>
> Using the Gaussian function to achieve uniformity has many advantages. The uniformity metric is asymptotically correct, and the uniformity metric is empirically reasonable with finite number of points. However, it has two problems. a) It relies on correct negative sampling, and b) it has a high time complexity of $O(n^2)$. In contrast, $L_{RT-FUG^-}$ avoids these drawbacks. $L_{RT-FUG^-}$ eliminates the need for manual negative sampling, thereby avoiding sampling bias. Additionally, the time complexity of $L_{RT-FUG^-}$ is $O(n)$. We will include relevant explanations in future versions.
>
> **Lastly,** thanks for your comments. We would be very grateful if you could engage in further discussions with us. This would be highly beneficial for our work.

---

### Official Review · Reviewer_5FZS · 2024-07-11

**Soundness:** 3
**Presentation:** 3
**Contribution:** 3
**Rating:** 7
**Confidence:** 5

**Summary:**

In this paper, it is discovered that contrastive learning and PCA follow the same framework. By combining the two, the authors propose a graph pre-training model that offers both the generalization capability of PCA for arbitrary node shapes and the powerful embedding ability of GCLs. This approach enhances the generality of existing graph pre-training frameworks.

**Strengths:**

- The authors present a novel dimensional encoding neural network, which appears to be a simple yet effective approach for encoding global information.
- The paper innovatively unifies GCL with traditional PCA both theoretically and methodologically.
- Extensive experimental analysis is provided to validate the effectiveness of the proposed model.

**Weaknesses:**

- The primary advantage of the proposed model lies in its robust cross-domain encoding capability. However, the paper lacks a discussion of existing works on graph cross-domain learning [1].
- The authors claim that PCA is limited by its inability to encode homogeneous information within the data and the inflexibility of the encoder. However, they do not explain why they did not directly use a graph encoder on the PCA-generated embeddings.
- Equation 10 reduces the Euclidean distance between neighboring node representations. This implies forcing the alignment of neighbor features across each dimension, which seems to contradict the authors' idea of relative semantic encoding. It appears that a constraint on the similarity would better align with this idea.
- The proposed model seems to lack a projector between representations and the loss, a component widely present in GCLs. The authors should provide more discussion to explain this, or it may be that they omitted the introduction of it.

[1] Unsupervised Domain Adaptive Graph Convolutional Networks, WWW’ 20.

**Questions:**

- Table 2 demonstrates the remarkable transfer learning capabilities of FUG. However, does this indicate that collaborative training on multiple datasets does not significantly improve the model's performance in cross-domain scenarios? A similar observation is made in GCC, where models pre-trained on multiple datasets do not show a substantial performance improvement compared to those pre-trained on a single dataset.

**Limitations:**

Yes, they have.

---

> ### Author Rebuttal · Authors · 2024-08-06
>
> Thanks you very much for your comments. The following are our response.
>
> # R1 for W1.
> The works related to Graph Domain Adaptation (GDA) primarily focus on knowledge transfer between domains. They differ significantly from FUG in several ways.
> - First, GDAs are typically supervised in the source domain and often use data from the target domain during training. In contrast, graph pre-training models are mostly fully self-supervised or unsupervised.
> - Second, Existing GDAs use datasets with fixed feature shapes. For example, in the work you mentioned, the feature dimensions of the datasets are fixed at 7537. In reality, a trained GDA cannot be universally applied to diverse graph data.
>
> Thank you for your suggestion, we will supplement our discussion with GDA-related works in future versions.
>
> # R2 for W2.
> This is mainly due to two factors:
> - First,  as described in Section 1, PCA, as a classic machine learning algorithm, can only serve as a data preprocessing method and cannot participate in graph pre-training. Consequently, during data preprocessing, PCA loses a substantial amount of important information that cannot be recovered through pre-training.
> - Second, another disadvantage of PCA methods is their inability to transform features to arbitrary shapes. PCA can only perform dimension reduction, not dimension increase. The dimension of node features varies greatly, as shown in Table 7 (Appendix), where the feature dimension of Physics is 8415, while PubMed is only 500. Using PCA preprocessing can only reduce the dimension of Physics to 500 to match PubMed, resulting in severe information loss.
>
> We will add these discussions in the future versions.
>
> # R3 for W3.
> Admittedly, using similarity can also align the relative semantics between positives. However, using the l2 distance has advantages, mainly in that.
> - First, using the l2 distance can better balance the gradients. The other two losses are defined based on the l2 distance, so defining $L_{RT-FUG+}$ using the l2 distance can make the values of the three losses more comparable, thereby balancing the gradients.
>
> - Second, the l2 distance and similarity are equivalent. As mentioned in Section 3 and Appendix A, reducing the l2 distance and increasing similarity are equivalent under the assumption that the length of the representation vectors is the same. Additionally, both $L_{DE}$ and $L_{RT-FUG-}$ include normalization operations, which essentially act as flexible similarity constraints between representations. The l2 distance constraint in $L_{RT-FUG+}$ can indirectly ensure that the representations satisfy the assumption of equal lengths.
>
> We will add relevant discussions in future versions.
>
> # R4 for W4.
> FUG does not require a projector, demonstrating the effectiveness of the proxy tasks designed for FUG. We believe that the existing of projectors prevents overfitting to the proxy tasks, thereby increasing the quality of representations. Essentially, the overfitting problem stems from the proxy tasks, as existing CL losses bring bias from negative sampling. However, $L_{RT-FUG-}$ avoids negative sampling by utilizing the concept of global uniformity, thus avoiding artificially defined negative samples. Consequently, our proxy task is more effective and can directly constrain representations, leading to better model performance.
>
> # R5 for Q1.
> This is a very valuable question. As you mentioned, like GCC, multi-datasets collaborative training only yields slight improvements for FUG. As stated in Section 5.2, we believe this is mainly because the proposed RT task is easy to learn and can be generalized across datasets. Therefore, a small amount of data training can achieve performance close to that in intra-domain scenarios. Additionally, this might be because existing tasks like node classification and link prediction are relatively easy, and the information quality provided by the datasets is high, facilitating knowledge transfer. We will attempt to challenge more complex graph scenarios in future works to explore more effective graph pre-training models.

---

> > ### Comment · Reviewer_5FZS · 2024-08-12
> >
> > Thanks for your rebuttal. I would keep the original score.

---

> > > ### Author Response · Authors · 2024-08-12
> > >
> > > Thank you again for your review and the score. We will further improve our paper follow your suggestions.

---

### Official Review · Reviewer_FBBq · 2024-07-11

**Soundness:** 3
**Presentation:** 3
**Contribution:** 3
**Rating:** 7
**Confidence:** 4

**Summary:**

The paper is generally well-presented. It aims to address the generalizability of graph pre-training models to graph data with arbitrary node feature shapes. By comparing the Principal Component Analysis (PCA) with existing graph contrastive learning methods, the authors discovered that PCA follows the same framework as these graph contrastive learning methods. Based on this observation, they proposed a Feature-Universal Graph Pre-training (FUG) strategy and instantiated a FUG model. Extensive experiments have been conducted to validate the effectiveness of the proposed method.

**Strengths:**

1. The proposed FUG strategy is novel and significant. Pre-trained graph models struggle to generalize across various graph datasets. The FUG strategy introduces the concept of dimensional encoding, which nearly losslessly unifies features to the same dimension. This innovative and interesting approach can provide new insights for researchers.
2. The idea of relative semantic encoding is interesting, as it unifies the optimization objectives of contrastive learning and PCA. Additionally, this objective shows potential in learning domain-invariant knowledge. The instantiated loss function is effective and efficient.
3. The authors conduct a theoretical analysis showing that classical PCA and contrastive learning fundamentally follow the same theoretical framework。
4. The proposed FUG model demonstrates significant performance in cross-domain learning experiments.

**Weaknesses:**

1. Equation 4 is somewhat unclear. Although the authors provide an intuitive explanation, a more effective approach would be to formally decompose PCA according to the form on the right side of the equation, making it easier to understand.
2. The authors state that FUG is inspired by existing works discussing the relationship between PCA and contrastive learning. The authors should more clearly introduce these works and compare them with the proposed theory to highlight the theoretical contributions.

Minor issues and typos:
1. The subscript font for symbols should be consistently either italic or upright. Additionally, Table 5 should specify the units of the values.
2. Line 180: Maximize $[Sim(x_i, x_j )$

**Questions:**

Please see the weaknesses.

**Limitations:**

Please see the weaknesses.

---

> ### Author Rebuttal · Authors · 2024-08-06
>
> Thanks for your comments, which are incredibly helpful for improving our paper.
>
> # R1 for W1.
> We apologize for not explaining this clearly. Here, we provide a formal definition of $Dim(\cdot)$ and $Fea(\cdot)$. $Dim(\cdot)$ is defined as a dimension-specific encoder, formalized as $DimEmb_i = Encode(X_{:,i}^T)$, and $Fea(\cdot)$ is defined as a sample-specific encoder, formalized as $FeaEmb_j = Encode(X_j, \epsilon)$, where $Encode$ represents a linear or nonlinear mapping, $\epsilon$ represents additional inputs. With these definitions, PCA can be decomposed as follows:
> $$DimEmb = CovEncode(\widetilde{X}^T) =\widetilde{X}^T *\widetilde{X}= \widetilde{X}^TW_1$$
> $$FeaEmb = Encode(\widetilde{X})=W_2 X, W_2 = \varphi(DimEmb)$$
> where $\widetilde{X}$ is the standardized feature matrix, $W_2$ is the parameter matrix for $Fea(\cdot)$, and $\varphi(DimEmb)$ represents the process of solving $W_2$, with $\varphi$ denoting matrix decomposition, sorting and concatenating based on eigenvalues. This clearly demonstrates how the PCA method can be decomposed into the $Dim(\cdot)$ and $Fea(\cdot)$ components.
>
> More detailed definitions and prove can be found in global rebuttal. We will incorporate this into the theoretical section to improve our paper.
>
> # R2 for W2.
> As mentioned in Sections 1 and 3, our work is primarily inspired by [1]. In [1], the InfoNCE loss is briefly explored as a generalized nonlinear PCA objective function. We further propose a relative semantic relationship encoding objective to better unify the contrastive loss and the PCA objective, and we provide additional analysis of the positive and negative parts. Additionally, as research for the CL and PCA frameworks, we extra-analyze aspects such as data augmentation, encoder, and sampling strategy. We will add these discussions in future versions.
>
> # R3 for W3.
> Thank you very much for your suggestions. We will correct them in future versions.
>
> [1] The trade-off between universality and label efficiency of representations from contrastive learning, ICLR 2023.

---

> > ### Comment · Reviewer_FBBq · 2024-08-10
> > **Response to authors**
> >
> > Thank you for your response. I would like to maintain my previous scores. By the way, please revise them carefully in your future version.

---

> > > ### Author Response · Authors · 2024-08-11
> > >
> > > We sincerely thank you for your recognition of our work and these valuable suggestions! We will carefully fix these issues in future versions and double-check our paper to improve its quality.

---

### Official Review · Reviewer_nhsa · 2024-07-12

**Soundness:** 3
**Presentation:** 3
**Contribution:** 4
**Rating:** 7
**Confidence:** 4

**Summary:**

The authors propose a universal graph pretraining framework called FUG. Inspired by PCA, FUG includes a parameterized dimension encoding component to unify different forms of node attributes, thereby preventing the loss of important semantics. The authors design an optimization objective based on relative semantics to learn invariant knowledge across datasets. FUG demonstrates excellent performance in cross-domain training.

**Strengths:**

1. This work is highly contributive. It has the potential to positively impact the broad field of graph representation learning, as it proposes a pretraining framework that is universally applicable to graph data.
2. The paper is well-written and easy to understand.
3. The authors' theoretical foundation is solid, offering a new perspective on traditional dimensionality reduction methods and advanced graph contrastive learning.

**Weaknesses:**

1. The authors introduce a new loss, which, in my view, has a primary advantage over the InfoNCE in terms of faster training speed. However, its effectiveness on positive sampling compared to existing GCLs is questionable as noise based augmentation not only aligns positive samples but also enhances the model's robustness. Using neighbors as positives might make the model sensitive to noise.
2. Although this paper is well-written, I still recommend that the authors emphasize the two questions raised in Section 3. This would enable readers to quickly find the questions corresponding to the two answers.
3. In Theorem 3.1, dividing PCA into Dim. and Fea. seems inappropriate. In fact, I tend to believe that PCA cannot be clearly separated into these two parts. Matrix decomposition belongs to Fea., but the ordering of eigenvalues might be more reasonably viewed as Dim., as it can be considered a nonlinear mapping based on the relative relationships between dimensions.

**Questions:**

Please see the weakness.

**Limitations:**

Yes

---

> ### Author Rebuttal · Authors · 2024-08-06
>
> Thank you very much for recognizing our work. Our responses are blow.
>
> # R1 for W1.
> The robustness improvement brought by data augmentation mainly stems from its ability to encode perturbed data into the same representation as clean data. Although FUG does not use data augmentation, it retains this characteristic through positive neighbor sampling. Specifically, in homogeneous graphs, linked nodes are considered to have the same semantics, even if their features and topologies are not entirely identical. Therefore, the differences between linked nodes can be seen as a form of perturbation. Compared to data augmentation, this is a method that does not require hyper-parameter tuning and adaptively adds noise, allowing it to ignore noise without encoding inter-class nodes into the same representation.
>
> # R2 for W2.
> Thanks for your suggestions. We will add them in future versions.
>
> # R3 for W3.
> Indeed, as a classic and effective machine learning algorithm, it is challenging to completely separate PCA into two unrelated parts. We have added a more precise definition of $Dim(\cdot)$ and $Fea(\cdot)$, where $Dim(\cdot)$ is formalized as $DimEmb_i = Encode(X_{:,i}^T)$, and $Fea(\cdot)$ is formalized as $FeaEmb_j = Encode(X_j, \epsilon)$, where $Encode$ represents a linear or nonlinear mapping and  $\epsilon$ represents additional inputs (details can be found in global rebuttal). Under this definition, PCA can be decomposed into $Dim(\cdot)$ and $Fea(\cdot)$ as described in Section 3, because the ordering of eigenvalues can be seen as the process of solving the weights of the $Encode$ in $Fea(\cdot)$. We will include this explanation in future versions.

---

> > ### Comment · Reviewer_nhsa · 2024-08-12
> > **About the responses**
> >
> > Thank you for the response, which well addressed my main concerns. I have carefully read the response and the other reviews. In my opinion, the authors proposed a highly universal graph pre-training strategy, supported by theoretical and experimental evidence, which is both sound and interesting to me. So, I support the acceptance of this paper.

---

### Official Review · Reviewer_vBXZ · 2024-07-12

**Soundness:** 3
**Presentation:** 3
**Contribution:** 2
**Rating:** 5
**Confidence:** 4

**Summary:**

In this paper, the authors try to answer the question: Could a graph contrastive pre-training principle based on PCA theory be developed to address inherent issues of PCA and broadly apply to node features of different shapes? Then the authors start with some theoretical analysis and then propose a method namely FUG, which has good transferability. The method is validated on several datasets to test the performance on cross-domain and inter-domain tasks.

**Strengths:**

-   This paper is pretty well-written and easy to follow.
-   The paper is well-motivated. It is motivated by the connection between PCA and Contrastive Learning since PCA has good transferability.
-   A different contrastive loss is proposed to replace InfoNCE.
-   The ablation experiments show some interesting conclusions.

**Weaknesses:**

-   The theoretical analysis is somewhat loose. For instance, in Theorem 3.1, $Dim(\cdot)$ and $Fea(\cdot)$ are loose formulations since their mathematical properties are not well defined. The authors should clarify that a function or mapping is a Dim as long as it satisfies some pre-conditions. The theoretical analysis may not be qualified for NeurIPS. Moreover, the equivalence between $\mathcal{L}_{RT}$ and InfoNCE is quite similar to some existing works, such as [1]. So the contribution from the theoretical aspect may be over-claimed.
-   From the ablation experiments, it seems that CE is the most important part. However, it is unclear why CE can significantly improve the performance. A theoretical proof will be convincing.
-   The pre-training of GNNs has been studied recently. However, the proposed FUG is not compared with these methods.  Here is a survey as an example [2].
-   There are totally 3 hyperparameters in the final loss, which may result in great difficulty in tuning them in practice.

[1] A unifying mutual information view of metric learning: cross-entropy vs. pairwise losses

[2] Graph Prompt Learning: A Comprehensive Survey and Beyond

**Questions:**

-   What is the connection between the proposed FUG and graph prompt tuning? Which one is more practical in practice?
-   Why does CE improve performance significantly?
-   Can RT-FUG- and RT-FUG+ be replaced by InfoNCE or other contrastive loss?

For more questions, please see Weaknesses.

Overall, I'd like to discuss this paper with the authors and other reviewers during rebuttal. And then update my rating.

**Limitations:**

Yes

---

> ### Author Rebuttal · Authors · 2024-08-06
>
> Thanks for your thorough review of our paper and your comments. Here are our responses.
>
> # R1 for W1.
> (a) We first give the definition of $Dim(\cdot)$ and $Fea(\cdot)$.
>
> Given a data matrix $X\in\mathbb{R}^{N\times D}$, $N$ represents the number of samples and $D$ represents the number of dimensions.
>
> $Dim(\cdot)$ refers to a serious of dimension-specific encoders that directly accept multi-samples input with a specific dimension $X_{:,i}^T$ and output the embedding of that dimension, denoted as $DimEmb_i = Encode(X_{:,i}^T)$, $DimEmb\in \mathbb{R}^{D\times N’}$. In contrast, $Fea(\cdot)$ refers to a serious of sample-specific encoders that accept all dimensions of a specific sample as input and output the embedding of that sample, denoted as $FeaEmb_j = Encode(X_j, \epsilon)$, $FeaEmb\in\mathbb{R}^{N\times D’}$. Here, $Encode$ represent linear or non-linear functions, $\epsilon$ denotes additional inputs.
>
> With the clear definitions, PCA can be decomposed into $Dim(\cdot)$ and $Fea(\cdot)$, with a rigorous proof, which can be found in Global Rebuttal.
>
> (b) Secondly, although Ref [1] you provided decomposes InfoNCE into the tightness part and the contrastive part, our work is theoretically entirely different from it.
> - **Different Contributions.** In [1], the authors aim to analyzing the relationships between several CE-based losses, whereas we aim to unify the PCAs and CL frameworks. Work [1] might be somewhat similar to a small part (task analysis) of our theory. Apart from that, we extra-analyze the augmentation, encoder and sampling strategy.
> - **Relative Semantic Perspective.** In task analysis, we provided a new relative semantic perspective, whereas work [1] tried to connect InfoNCE with labels $y$ (as shown in Equations 3 and Table 2 of work [1]). Given that labels are unknown in SSL, our provided perspective, which only focus on the relationship between data, is more applicable.
> - **Positive Part Analyzing.** Work [1] only demonstrated that positive pairs in InfoNCE are aligned. In contrast, our work reveals that aligning augmented views of the same data inherently aligns representations between different data.
>
> We will increase these in future versions to refine the theory. We will also greatly hope to discuss the theoretical part with the reviewer to further improve this work.
> # R2 for W2 and Q2.
> We guess here you may refer to $L_{DE}$ (in Table 4) rather than CE since FUG does not include a CE component.
>
> The main reason of its significance is that, $L_{RT-FUG^+}$ and $L_{RT-FUG^-}$ are effective when based on $L_{DE}$, especially for datasets with high quality features. As described in Section 5.4, when $L_{DE}$ is absent, DE is not sufficiently trained, which leads to DE embedding $X$ almost randomly to generate $H$, which is an important input for GE. Furthermore, without the guidance of $L_{DE}$, GE only needs to extract relationships in $H$ to reduce the loss, ignoring whether $H$ contains sufficient information from $X$. Therefore, in FUG w/o $L_{DE}$, the $H$ input to GE is of poor quality, resulting in worst performances on some datasets. We will include more explanations in future versions to facilitate understanding.
>
> # R3 for W3 and Q1.
> Consider that Ref [2] you provided in W3 is also about Graph Prompt Learning, so we combine our responses to your W3 and Q1.
> - First, it seems that Prompt Learning (PL) and Pre-Training (PT) are two distinct directions for improving graph learning. To be specific, PT (that you provided) focuses on making models learn knowledge applicable to various data and generate representations, whereas PL (which our FUG belongs to) aims to generalize pre-trained models to different downstream tasks, including few-shot and zero-shot scenarios. Thus, as a universal graph pre-training model, our FUG is primarily compared to some advanced PT models in Tables 2 and 3. Notably, in Table 2, our FUG significantly outperforms some latest SOTA universal PT models, which demonstrates the effectiveness of FUG as a universal graph pre-training approach.
> - Second, PL and PT are not mutually exclusive while may work together to achieve better performance. As a flexible PT strategy, FUG can accept various graph inputs and can be integrated with different PLs, thereby extending their generalizability across diverse graph datasets. Additionally, integrating PLs also enhances FUG's performance in few-shot and zero-shot scenarios. Exploring the combination of FUG with various PLs will be our future research direction.
>
> # R4 for W4.
> In fact, the FUG model has only one more hyper-parameter compared to classical GCLs (or even fewer, as FUG does not require tuning multiple parameters related to augmentation).
> - First, in Equation 11, $\lambda_1$ is fixed at 1, requiring only two hyper-parameters to be tuned, while classic GCLs require tuning the temperature parameter $\tau$. Therefore, FUG has just one more hyper-parameter to be tuned.
> - Second, as shown in Table 9 (in the appendix), the search spaces for $\lambda_2$ and $\lambda_3$ are small, specifically [0.1, 0.3, 0.5] and [20, 50, 100, 200, 300], respectively. They exhibit clear unimodality in sensitive analysis in Figure 4, making the tuning process straightforward.
> - Furthermore, as shown in Table 2 and Figure 4, the model can achieve good results even without meticulous tuning, demonstrating the robustness of FUG.
>
> We will add these discussions in the future versions.
>
> # R5 for Q3.
> Yes, they can be replaced by other contrastive losses such as InfoNCE since FUG is a highly inclusive theory. However, compared to our losses, InfoNCE have some drawbacks. For example, as described in Section 3 and Appendix B, InfoNCE introduce negative sampling bias and has high computational complexity. These issues are critical for universal graph pre-training models.
>
> **Finally,** thank you again for your comments. We look forward to further discussing with you to address your concerns and to continually improve our paper.

---

> > ### Comment · Reviewer_vBXZ · 2024-08-12
> >
> > I thank the authors for the response. I have read the response and other reviews. I admit that the paper has some merits. I would raise my rating to 5.

---

> > > ### Author Response · Authors · 2024-08-12
> > >
> > > We sincerely appreciate the time and effort you invested in reviewing our paper, as well as your new score! We will carefully revise our paper according to your suggestions. Thank you again for your valuable comments, which have greatly enhanced the theoretical part of our work.

---

### Author Rebuttal · Authors · 2024-08-06

We sincerely thank the five reviewers for their constructive suggestions and comments, which have greatly helped us improve our paper. We also extend our gratitude to the NeurIPS Chairs for reviewing our paper.

We have noticed that some reviewers raised concerns regarding Theorem 3.1. In our paper, we provided an intuitive explanation of the two proposed encoders, $Dim(\cdot)$ and $Fea(\cdot)$, and offered a rigorous proof of the equivalence between the CL loss, the PCA objective, and the proposed RT task. Here, we present a clearer definition of $Dim(\cdot)$ and $Fea(\cdot)$ and a theoretical proof of how PCA can be decomposed into these two parts, to enhance our theoretical foundation.

Given a set of data $X \in \mathbb{R}^{N \times D}$, where $N$ represents the number of samples, and $D$ represents the number of dimensions. Let $Encode$ represent a series of linear or nonlinear mapping functions.

**Definition of $Dim(\cdot)$.** $Dim(\cdot)$ refers to a series of dimension-specific encoders that directly take the data of all samples under a specific dimension $i$, $X_{:,i}^T$, and output the embedding of that dimension, $DimEmb_i$, formalized as $DimEmb_i = Encode(X_{:,i}^T)$, $DimEmb \in \mathbb{R}^{D \times N'}$.

**Definition of $Fea(\cdot)$.** $Fea(\cdot)$ refers to a series of sample-specific encoders that take the data of all dimensions of a specific sample $j$, $X_{j}$, and output the embedding of that sample, $FeaEmb_j$, formalized as $FeaEmb_j = Encode(X_j, \epsilon)$, $FeaEmb \in \mathbb{R}^{N \times D'}$, where $\epsilon $ represents additional input.

Based on the above definitions, PCA can be decomposed into $Fea(\cdot)$ and $Dim(\cdot)$ parts, with a detailed proof as follows:

> Given an input data $X$ with a standardized matrix $\widetilde{X}=X-Mean(X)$, PCA is as follows.
> $$PCA(X) = \widetilde{X}P, P=Norm[Concat(c_i|c_i\in C, \lambda_c \ge \lambda^{(k)})], (\Lambda, C) = \delta\[Cov(\widetilde{X})],$$
where $(\Lambda, C)$ represent the eigenvalues and eigenvectors matrix, $Norm(\cdot)$ represents normalization, $Concat(\cdot)$ represents concatenation, $\lambda^{(k)}$ represents the k-th largest eigenvalue, $Cov(\cdot)$ represents covariance computation, and $\delta(\cdot)$ represents eigenvalue and eigenvector computation. This can be decomposed into the forms of $Dim(\cdot)$ and $Fea(\cdot)$ as follows:
> $$ DimEmb =  Encode_1(\widetilde{X}) = \widetilde{X}^TW_1, W_1=\widetilde{X},$$
> $$FeaEmb = Encode_2( \widetilde{X}) = PCA(X) = \widetilde{X}W_2, W_2 = P= \varphi(DimEmb),$$
> where $W_1$ is a special parameter matrix, $W_1=\widetilde{X}$, and $\varphi(\cdot)$ represents the operation to solve $W_2$ as follows.
> $$W_2 = P = \varphi(DimEmb) = Norm[Concat(c_i|c_i\in C, \lambda_c \ge \lambda^{(k)})], (\Lambda, C) = \delta(DimEmb).$$
> From the above, $Encode_1(\cdot)$ and $Encode_2(\cdot)$ satisfy linear or nonlinear mappings. Therefore, we derive:
> $$PCA(X) = \widetilde{X}P = \widetilde{X}W_2 = \widetilde{X} \delta(Dim(\widetilde{X})) = Fea(\widetilde{X}, Dim(\widetilde{X})).$$
> Thus, PCA can be rigorously decomposed into $Dim(\cdot)$ and $Fea(\cdot)$ parts as described in Theorem 3.1.

We will include the above content in future versions of this paper to further support our theory. Additionally, we have provided relevant content and detailed discussion in each reviewer's response.

Once again, we thank all the reviewers. We have carefully read all the comments and made every effort to address all concerns. We hope that the rebuttal phase can be informative and pleasant for all reviewers.

---

### Decision · Program_Chairs · 2024-09-25

**Decision:**

Accept (poster)

**Comment:**

The authors introduce Feature-Universal Graph Pre-training (FUG), a novel framework inspired by Principal Component Analysis (PCA) designed to enhance the generalizability of graph pre-training models. FUG features a parameterized dimension encoding component that unifies various node attributes while preserving essential semantics. By integrating the principles of PCA and Graph Contrastive Learning (GCL), FUG aims to learn invariant knowledge across different datasets. The model's innovative approach addresses the challenge of generalizing to graph data with diverse node feature shapes, and extensive experiments demonstrate its effectiveness, particularly in cross-domain training scenarios.

The reviewers praise the paper's clarity, solid theoretical foundation, and innovative approach that unifies Principal Component Analysis (PCA) with Graph Contrastive Learning (GCL) both methodologically and theoretically. The Feature-Universal Graph Pre-training (FUG) strategy is recognized as a significant contribution to the field of graph representation learning. While some issues were raised concerning the theoretical analyses and experimental evaluations, the reviewers were generally satisfied with the authors' further explanations provided during the discussion period.